# Decomposed Diffusion Sampler for Accelerating Large-Scale Inverse Problems

**Hyungjin Chung[1], Suhyeon Lee[2], Jong Chul Ye[2]**
[1] Dept. of Bio & Brain Engineering, KAIST,   [2] Kim Jae Chul Graduate School of AI, KAIST
{hj.chung, suhyeon.lee, jong.ye}@kaist.ac.kr

## Abstract

Krylov subspace, which is generated by multiplying a given vector by the matrix of a linear transformation and its successive powers, has been extensively studied in classical optimization literature to design algorithms that converge quickly for large linear inverse problems. For example, the conjugate gradient method (CG), one of the most popular Krylov subspace methods, is based on the idea of minimizing the residual error in the Krylov subspace. However, with the recent advancement of high-performance diffusion solvers for inverse problems, it is not clear how classical wisdom can be synergistically combined with modern diffusion models. In this study, we propose a novel and efficient diffusion sampling strategy that synergistically combines the diffusion sampling and Krylov subspace methods. Specifically, we prove that if the tangent space at a denoised sample by Tweedie's formula forms a Krylov subspace, then the CG initialized with the denoised data ensures the data consistency update to remain in the tangent space. This negates the need to compute the manifold-constrained gradient (MCG), leading to a more efficient diffusion sampling method. Our method is applicable regardless of the parametrization and setting (i.e., VE, VP). Notably, we achieve state-of-the-art reconstruction quality on challenging real-world medical inverse imaging problems, including multi-coil MRI reconstruction and 3D CT reconstruction. Moreover, our proposed method achieves more than 80 times faster inference time than the previous state-of-the-art method. Code is available at `https://github.com/HJ-harry/DDS`

## 1 Introduction

Diffusion models (Ho et al., 2020; Song et al., 2021b) are the state-of-the-art generative model that learns to generate data by gradual denoising, starting from the reference distribution (e.g. Gaussian). In addition to superior sample quality in the unconditional sampling of the data distribution, it was shown that one can use unconditional diffusion models as *generative priors* that model the data distribution (Chung et al., 2022a; Kawar et al., 2022), and incorporate the information from the forward physics model along with the measurement $\boldsymbol{y}$ to sample from the posterior distribution $p(\boldsymbol{x}|\boldsymbol{y})$. This property is especially intriguing when seen in the context of Bayesian inference, as we can separate the parameterized prior $p_\theta(\boldsymbol{x})$ from the measurement model (i.e. likelihood) $p(\boldsymbol{y}|\boldsymbol{x})$ to construct the posterior $p_\theta(\boldsymbol{x}|\boldsymbol{y}) \propto p_\theta(\boldsymbol{x})p(\boldsymbol{y}|\boldsymbol{x})$. In other words, one can use the same pre-trained neural network model *regardless* of the forward model at hand. Throughout the manuscript, we refer to this class of methods as **D**iffusion model-based **I**nverse problem **S**olvers (DIS).

For inverse problems in medical imaging, e.g. magnetic resonance imaging (MRI), computed tomography (CT), it is often required to accelerate the measurement process by reducing the number of measurements. However, the data acquisition scheme may vary vastly according to the circumstances (e.g. vendor, sequence, etc.), and hence the reconstruction algorithm needs to be adaptable to different possibilities. Supervised learning schemes show weakness in this aspect as they overfit to the measurement types that were used during training. As such, it is easy to see that diffusion models will be particularly strong in this aspect as they are agnostic to the different forward models at inference. Indeed, it was shown in some of the pioneering works that diffusion-based reconstruction algorithms have high generalizability (Jalal et al., 2021; Chung & Ye, 2022; Song et al., 2022; Chung et al., 2023b).

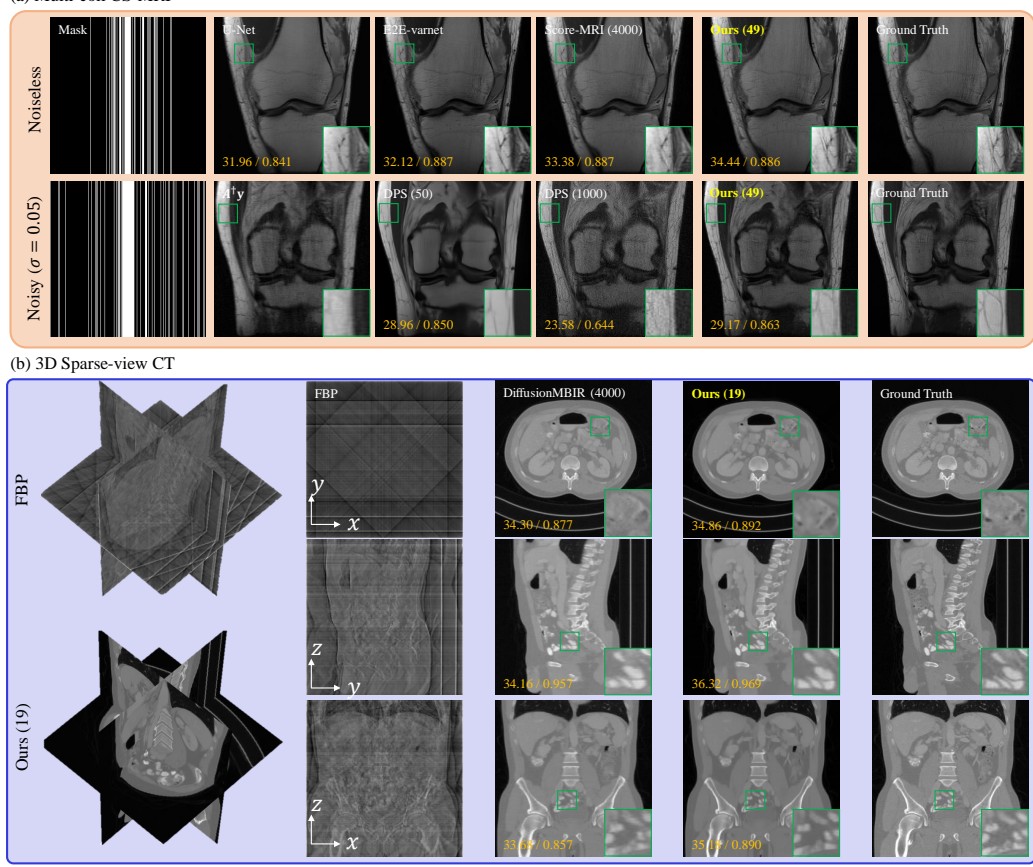

Figure 2: Representative reconstruction results. (a) Multi-coil MRI reconstruction, (b) 3D sparse-view CT. Numbers in parenthesis: NFE. Yellow numbers in bottom left corner: PSNR/SSIM.

While showing superiority in performance and generalization capacity, slow inference time is known to be a critical drawback of diffusion models. One of the most widely acknowledged accelerated diffusion sampling strategies is the denoising diffusion implicit model (DDIM) (Song et al., 2021a), where the stochastic ancestral sampling of denoising diffusion probabilistic models (DDPM) can be transitioned to deterministic sampling, and thereby accelerate the sampling. Accordingly, DDIM sampling has been well incorporated in solving low-level vision inverse problems (Kawar et al., 2022; Song et al., 2023). In a recent application of DDIM for linear image restoration tasks,

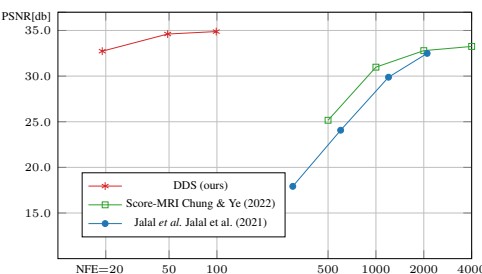

Figure 1: Parallel imaging MR reconstruction evaluation PSNR vs. NFE (log scale). Reconstruction from 1D uniform random ×4 acceleration (Zbontar et al., 2018).

Wang et al. (2023) proposed an algorithm dubbed denoising diffusion null-space models (DDNM), where one-step null-space modification is made to impose consistency. However, the sampling strategy is not successful in practical large-scale medical imaging contexts when the forward model is significantly more complex (e.g. parallel imaging (PI) CS-MRI, 3D modalities). Furthermore, it is unclear how the algorithm is related to the existing literature of conditional sampling approaches that take into account the geometry of the manifold (Chung et al., 2022a; 2023a).

On the other hand, in classical optimization literature, Krylov subspace methods have been extensively studied to deal with large-scale inverse problems due to their rapid convergence rates (Liesen &

Strakos, 2013). Specifically, consider a typical linear inverse problem

$$y = Ax, \tag{1}$$

where $A$ is the linear mapping and the goal is to retrieve $x$ from the measurement $y$. Without loss of generality, throughout the paper we assume that $A$ in (1) is square. Otherwise, we can obtain an equivalent inverse problem with symmetric linear mapping $\tilde{A}$ as $\tilde{y} := A^*y = A^*Ax := \tilde{A}x$. This is because the solution to the normal equation $A^*Ax = A^*y$ is indeed a solution to $Ax = y$ if $A^*$ has full column rank, which holds in most of the ill-posed inverse problem cases. Given an initial guess $\hat{x}$, Krylov subspace methods seek an approximate solution $x^{(l)}$ from an affine subspace $\hat{x} + \mathcal{K}_l$, where the $l$-th order Krylov subspace $\mathcal{K}_l$ is defined by

$$\mathcal{K}_l := \mathrm{Span}(b, Ab, \cdots, A^{l-1}b), \quad b := y - A\hat{x} \tag{2}$$

For example, the conjugate gradient (CG) method is a special class of the Krylov subspace method that minimizes the residual in the Krylov subspace. Krylov subspace methods are particularly useful for large-scale problems thanks to their fast convergence (Liesen & Strakos, 2013).

Inspired by this, here we are interested in developing a method that synergistically combines Krylov subspace methods with diffusion models such that it can be effectively used for large-scale inverse problems. Specifically, based on a novel observation that a diffusion posterior sampling (DPS) (Chung et al., 2023a) with the manifold constrained gradient (MCG) (Chung et al., 2022a) is equivalent to one-step projected gradient on the tangent space at the "denoised" data by Tweedie's formula, we provide *multi-step* update scheme on the tangent space using Krylov subspace methods. Specifically, we show that the multiple CG updates are guaranteed to remain in the tangent space, and subsequently generated sample with the addition of the noise component can be correctly transferred to the next noisy manifold. This eliminates the need for the computationally demanding MCG while permitting multiple *economical* CG steps at each ancestral diffusion sampling, resulting in a more efficient DDIM sampling. Our analysis holds for both variance-preserving (VP) and variance-exploding (VE) sampling schemes.

The combined strategy, dubbed **D**ecomposed **D**iffusion **S**ampling (DDS), yields *better* performance with much-reduced sampling time (20~50 NFE, $\times 80 \sim 200$ acceleration; See Fig. 1 for comparison, Fig. 2 for representative results), and is shown to be applicable to a variety of challenging large scale inverse problem tasks: multi-coil MRI reconstruction and 3D CT reconstruction.

## 2 BACKGROUND

**Krylov subspace methods** Consider the linear system (1). In classical projection based methods such as Krylov subspace methods (Liesen & Strakos, 2013), for given two subspace $\mathcal{K}$ and $\mathcal{L}$, we define an approximate problem to find $x \in \mathcal{K}$ such that

$$y - Ax \perp \mathcal{L} \tag{3}$$

This is a basic projection step, and the sequence of such steps is applied. For example, with non-zero estimate $\hat{x}$, the associated problem is to find $x \in \hat{x} + \mathcal{K}$ such that $y - Ax \perp \mathcal{L}$, which is equivalent to finding $\delta \in \mathcal{K}$ such that

$$b - A\delta \perp \mathcal{L}, \quad \delta := x - \hat{x}, \quad b := y - A\hat{x}. \tag{4}$$

In terms of the choice of the two subspaces, the CG method chooses the two subspaces $\mathcal{K}$ and $\mathcal{L}$ as the same Krylov subspace:

$$\mathcal{K} = \mathcal{L} = \mathcal{K}_l := \mathrm{Span}(b, Ab, \cdots, A^{l-1}b). \tag{5}$$

Then, CG attempts to find the solution to the following optimization problem:

$$\min_{x \in \hat{x} + \mathcal{K}_l} \|y - Ax\|^2 \tag{6}$$

Krylov subspace methods can be also extended to nonlinear problems via zero-finding. Specifically, the optimization problem $\min_x \ell(x)$ can be equivalently converted to a zero-finding problem of its gradient, i.e. $\nabla_x \ell(x) = 0$. If we consider a non-linear forward imaging operator $\mathcal{A}(\cdot)$, we can define $\ell(x) = \|y - \mathcal{A}(x)\|^2/2$. Then, one can use, for example, Newton-Krylov method (Knoll & Keyes,

2004) to linearize the problem near the current solution and apply standard Krylov methods to solve the current problem. Now, given the optimization problem, we can see the fundamental differences between the gradient-based approaches and Krylov methods. Specifically, gradient methods are based on the iteration:

$$\boldsymbol{x}^{(i+1)} = \boldsymbol{x}^{(i)} - \gamma \nabla_{\boldsymbol{x}} \ell(\boldsymbol{x}^{(i)}) \tag{7}$$

which stops updating when $\nabla_{\boldsymbol{x}} \ell(\boldsymbol{x}^{(i)}) \simeq \boldsymbol{0}$. On the other hand, Krylov subspace methods try to find $\boldsymbol{x} \in \mathcal{K}_l$ by increasing $l$ to achieve a better approximation of $\nabla_{\boldsymbol{x}} \ell(\boldsymbol{x}) = \boldsymbol{0}$. This difference allows us to devise a computationally efficient algorithm when combined with diffusion models, which we investigate in this paper. See Appendix A.2 fur further mathematical background.

**Diffusion models** Diffusion models (Ho et al., 2020) attempt to model the data distribution $p_{\text{data}}(\boldsymbol{x})$ by constructing a hierarchical latent variable model

$$p_{\theta}(\boldsymbol{x}_0) = \int p_{\theta}(\boldsymbol{x}_T) \prod_{t=1}^{T} p_{\theta}^{(t)}(\boldsymbol{x}_{t-1}|\boldsymbol{x}_t) \, d\boldsymbol{x}_{1:T}, \tag{8}$$

where $\boldsymbol{x}_{\{1,\dots,T\}} \in \mathbb{R}^d$ are *noisy* latent variables that have the same dimension as the data random vector $\boldsymbol{x}_0 \in \mathbb{R}^d$, defined by the Markovian forward conditional densities

$$q(\boldsymbol{x}_t|\boldsymbol{x}_{t-1}) = \mathcal{N}(\boldsymbol{x}_t|\sqrt{\beta_t}\boldsymbol{x}_{t-1}, (1-\beta_t)I), \tag{9}$$

$$q(\boldsymbol{x}_t|\boldsymbol{x}_0) = \mathcal{N}(\boldsymbol{x}_t|\sqrt{\bar{\alpha}_t}\boldsymbol{x}_0, (1-\bar{\alpha}_t)I). \tag{10}$$

Here, the noise schedule $\beta_t$ is an increasing sequence of $t$, with $\bar{\alpha}_t := \prod_{i=1}^{t} \alpha_t$, $\alpha_t := 1 - \beta_t$. Training of diffusion models amounts to training a multi-noise level residual denoiser (i.e. epsilon matching)

$$\min_{\theta} \mathbb{E}_{\boldsymbol{x}_t \sim q(\boldsymbol{x}_t|\boldsymbol{x}_0), \boldsymbol{x}_0 \sim p_{\text{data}}(\boldsymbol{x}_0), \boldsymbol{\epsilon} \sim \mathcal{N}(0, \boldsymbol{I})} \left[ \|\boldsymbol{\epsilon}_{\theta}^{(t)}(\boldsymbol{x}_t) - \boldsymbol{\epsilon}\|_2^2 \right],$$

such that $\boldsymbol{\epsilon}_{\theta^*}^{(t)}(\boldsymbol{x}_t) \simeq \frac{\boldsymbol{x}_t - \sqrt{\bar{\alpha}_t}\boldsymbol{x}_0}{\sqrt{1-\bar{\alpha}_t}}$. Furthermore, it can be shown that epsilon matching is equivalent to the denoising score matching (DSM) (Hyvärinen & Dayan, 2005; Song & Ermon, 2019) objective up to a constant with different parameterization

$$\min_{\theta} \mathbb{E}_{\boldsymbol{x}_t, \boldsymbol{x}_0, \boldsymbol{\epsilon}} \left[ \|\boldsymbol{s}_{\theta}^{(t)}(\boldsymbol{x}_t) - \nabla_{\boldsymbol{x}_t} \log q(\boldsymbol{x}_t|\boldsymbol{x}_0)\|_2^2 \right], \tag{11}$$

such that $\boldsymbol{s}_{\theta^*}^{(t)}(\boldsymbol{x}_t) \simeq -\frac{\boldsymbol{x}_t - \sqrt{\bar{\alpha}_t}\boldsymbol{x}_0}{1-\bar{\alpha}_t} = -\boldsymbol{\epsilon}_{\theta^*}^{(t)}(\boldsymbol{x}_t)/\sqrt{1-\bar{\alpha}_t}$. For notational simplicity, we often denote $\hat{\boldsymbol{s}}_t, \hat{\boldsymbol{\epsilon}}_t, \hat{\boldsymbol{x}}_t$ instead of $\boldsymbol{s}_{\theta^*}^{(t)}(\boldsymbol{x}_t), \boldsymbol{\epsilon}_{\theta^*}^{(t)}(\boldsymbol{x}_t), \boldsymbol{x}_{\theta^*}^{(t)}(\boldsymbol{x}_t)$, representing the *estimated* score, noise, and noiseless image, respectively. Sampling from (8) can be implemented by ancestral sampling, which iteratively performs

$$\boldsymbol{x}_{t-1} = \frac{1}{\sqrt{\alpha_t}} \left( \boldsymbol{x}_t - \frac{1-\alpha_t}{\sqrt{1-\bar{\alpha}_t}} \hat{\boldsymbol{\epsilon}}_t \right) + \tilde{\beta}_t \boldsymbol{\epsilon}, \tag{12}$$

where $\tilde{\beta}_t := \frac{1-\bar{\alpha}_{t-1}}{1-\bar{\alpha}_t} \beta_t$.

One can also view ancestral sampling (12) as solving the reverse generative stochastic differential equation (SDE) of the variance preserving (VP) linear forward SDE (Song et al., 2021b). Additionally, one can construct the variance exploding (VE) SDE by setting $q(\boldsymbol{x}_t|\boldsymbol{x}_0) = \mathcal{N}(\boldsymbol{x}_t|\boldsymbol{x}_0, \sigma_t^2 \boldsymbol{I})$, which is in a form of Brownian motion. In Appendix A.1 we further review the background on diffusion models under the score/SDE perspective.

**DDIM** Seen either from the variational or the SDE perspective, diffusion models are inevitably slow to sample from. To overcome this issue, DDIM (Song et al., 2021a) proposes another method of sampling which only requires matching the marginal distributions $q(\boldsymbol{x}_t|\boldsymbol{x}_0)$. Specifically, the update rule is given as follows

$$\begin{aligned}
\boldsymbol{x}_{t-1} &= \sqrt{\bar{\alpha}_{t-1}}\hat{\boldsymbol{x}}_t + \sqrt{1-\bar{\alpha}_{t-1} - \eta^2 \tilde{\beta}_t^2} \boldsymbol{\epsilon}_{\theta^*}^{(t)}(\boldsymbol{x}_t) + \eta \tilde{\beta}_t \boldsymbol{\epsilon}, \\
&= \sqrt{\bar{\alpha}_{t-1}}\hat{\boldsymbol{x}}_t + \tilde{\boldsymbol{w}}_t
\end{aligned} \tag{13}$$

where $\hat{x}_t$ is the *denoised* estimate

$$\hat{x}_t := x_{\theta^*}^{(t)}(x_t) := \frac{1}{\sqrt{\bar{\alpha}_t}}(x_t - \sqrt{1 - \bar{\alpha}_t}\epsilon_{\theta^*}^{(t)}(x_t)), \tag{14}$$

which can also be equivalently derived from Tweedie's formula Efron (2011), and $\tilde{w}_t$ denotes the total noise given by

$$\tilde{w}_t := \sqrt{1 - \bar{\alpha}_{t-1} - \eta^2 \tilde{\beta}_t^2}\,\epsilon_{\theta^*}^{(t)}(x_t) + \eta\tilde{\beta}_t\epsilon \tag{15}$$

In (13), $\eta \in [0, 1]$ is a parameter controlling the stochasticity of the update rule: $\eta = 0.0$ leads to fully deterministic sampling, whereas $\eta = 1.0$ with $\tilde{\beta}_t = \sqrt{(1 - \bar{\alpha}_{t-1})/(1 - \bar{\alpha}_t)}\sqrt{1 - \bar{\alpha}_t/\bar{\alpha}_{t-1}}$ recovers the ancestral sampling of DDPMs.

It is important to note that the noise component $\tilde{w}_t$ properly matches the forward marginal (Song et al., 2021a). The direction $\tilde{w}_t$ of this transition is determined by the vector sum of the deterministic and the stochastic directional component. Accordingly, assuming optimality of $\epsilon_{\theta^*}^{(t)}$, the total noise $\tilde{w}_t$ in (15) can be represented by

$$\tilde{w}_t = \sqrt{1 - \bar{\alpha}_{t-1}}\tilde{\epsilon} \tag{16}$$

for some $\tilde{\epsilon} \sim \mathcal{N}(0, I)$ (see Appendix B). In other words, (13) is equivalently represented by $x_{t-1} = \sqrt{\bar{\alpha}_{t-1}}\hat{x}_t + \sqrt{1 - \bar{\alpha}_{t-1}}\tilde{\epsilon}$ for some $\tilde{\epsilon} \sim \mathcal{N}(0, I)$. Therefore, it can be seen that the difference between DDIM and DDPM lies only in the degree of dependence on the deterministic estimate of the noise component with feasible intermediate values $\eta \in (0, 1)$.

## 3 DECOMPOSED DIFFUSION SAMPLING

**Conditional diffusion for inverse problems**  The conditional diffusion sampling for inverse problems (Kawar et al., 2022; Chung et al., 2023a;b) attempts to solve the following optimization problem:

$$\min_{x \in \mathcal{M}} \ell(x) \tag{17}$$

where $\ell(x)$ denotes the data consistency (DC) loss (i.e., $\ell(x) = \|y - Ax\|^2/2$ for linear inverse problems) and $\mathcal{M}$ represents the clean data manifold. Consequently, it is essential to navigate in a way that minimizes cost while also identifying the correct clean manifold. Accordingly, most of the approaches use standard reverse diffusion (e.g. (12)), alternated with an operation to minimize the DC loss.

Recently, Chung et al. (2023a) proposed DPS, where the updated estimate from the noisy sample $x_t \in \mathcal{M}_t$ is constrained to stay on the same noisy manifold $\mathcal{M}_t$. This is achieved by computing the MCG (Chung et al., 2022a) on a noisy sample $x_t \in \mathcal{M}_t$ as $\nabla_{x_t}^{mcg}\ell(x_t) := \nabla_{x_t}\ell(\hat{x}_t)$, where $\hat{x}_t$ is the denoised sample in (14) through Tweedie's formula. The resulting algorithm can be stated as follows:

$$x_{t-1} = \sqrt{\bar{\alpha}_{t-1}}\left(\hat{x}_t - \gamma_t\nabla_{x_t}\ell(\hat{x}_t)\right) + \tilde{w}_t \tag{18}$$

where $\gamma_t > 0$ denotes the step size. Since parameterized score function $\epsilon_{\theta^*}^{(t)}(x_t)$ is trained with samples supported on $\mathcal{M}_t$, $\epsilon_{\theta^*}^{(t)}$ shows good performance on denoising $x_t \sim \mathcal{M}_t$, allowing precise transition to $\mathcal{M}_{t-1}$. Therefore, by performing (18) from $t = T$ to $t = 0$, we can solve the optimization problem (17) with $x_0 \in \mathcal{M}$. Unfortunately, the computation of MCG for DPS requires computationally expensive backpropagation and is often unstable (Poole et al., 2022; Du et al., 2023).

**Key observation**  By applying the chain rule for the MCG term in (18), we have

$$\nabla_{x_t}\ell(\hat{x}_t) = \frac{\partial\hat{x}_t}{\partial x_t}\nabla_{\hat{x}_t}\ell(\hat{x}_t)$$

where we use the denominator layout for vector calculus. Since $\nabla_{\hat{x}_t}\ell(\hat{x}_t)$ is a standard gradient, the main complexity of the MCG arises from the Jacobian term $\frac{\partial\hat{x}_t}{\partial x_t}$.

In the following Proposition 1, we show that if the underlying clean manifold forms an affine subspace, then the Jacobian term $\frac{\partial\hat{x}_t}{\partial x_t}$ is indeed the orthogonal projection on the clean manifold up to a scale

factor. Note that the affine subspace assumption is widely used in the diffusion literature that has been used when 1) studying the possibility of score estimation and distribution recovery (Chen et al., 2023), 2) showing the possibility of signal recovery (Rout et al., 2023a;b), and the most relevantly, 3) showing the geometrical view of the clean and the noisy manifolds (Chung et al., 2022a). Although it is difficult to assume in practice that the clean manifold forms an affine subspace, it could be approximated by piece-wise linear regions represented by the tangent subspace at $\hat{\boldsymbol{x}}_t$. Therefore, Proposition 1 is still valid in that approximate regime.

**Proposition 1** (Manifold Constrained Gradient). *Suppose the clean data manifold $\mathcal{M}$ is represented as an affine subspace and assumes the uniform distribution on $\mathcal{M}$. Then,*

$$\frac{\partial \hat{\boldsymbol{x}}_t}{\partial \boldsymbol{x}_t} = \frac{1}{\sqrt{\bar{\alpha}_t}} \mathcal{P}_{\mathcal{M}} \tag{19}$$

$$\hat{\boldsymbol{x}}_t - \gamma_t \nabla_{\boldsymbol{x}_t} \ell(\hat{\boldsymbol{x}}_t) = \mathcal{P}_{\mathcal{M}} \left( \hat{\boldsymbol{x}}_t - \zeta_t \nabla_{\hat{\boldsymbol{x}}_t} \ell(\hat{\boldsymbol{x}}_t) \right) \tag{20}$$

*for some $\zeta_t > 0$, where $\mathcal{P}_{\mathcal{M}}$ denotes the orthogonal projection to $\mathcal{M}$.*

Now, (20) in Proposition 1 indicates that if the clean data manifold is an affine subspace, the DPS corresponds to the projected gradient on the clean manifold. Nonetheless, a notable limitation of MCG is its inefficient use of a single projected gradient step for each ancestral diffusion sampling. Motivated by this, we aim to explore extensions that allow computationally efficient multi-step optimization steps per each ancestral sampling.

Specifically, let $\mathcal{T}_t$ denote the tangent space on the clean manifold at a denoised sample $\hat{\boldsymbol{x}}_t$ in (14). Suppose, furthermore, that there exists the $l$-th order Krylov subspace:

$$\mathcal{K}_{t,l} := \mathrm{Span}(\boldsymbol{b}, \boldsymbol{A}\boldsymbol{b}, \cdots, \boldsymbol{A}^{l-1}\boldsymbol{b}), \quad \boldsymbol{b} := \boldsymbol{y} - \boldsymbol{A}\hat{\boldsymbol{x}}_t \tag{21}$$

such that

$$\mathcal{T}_t = \hat{\boldsymbol{x}}_t + \mathcal{K}_{t,l}$$

Then, using the property of CG in (6), it is easy to see that $M$-step CG update with $M \leq l$ starting from $\hat{\boldsymbol{x}}_t$ are confined in $\mathcal{T}_t$ since it corresponds to the solution of

$$\min_{\boldsymbol{x} \in \hat{\boldsymbol{x}}_t + \mathcal{K}_M} \|\boldsymbol{y} - \boldsymbol{A}\boldsymbol{x}\|^2 \tag{22}$$

and $\mathcal{K}_M \subset \mathcal{K}_l$ when $M \leq l$. This offers a pivotal insight. It shows that if the tangent space at each denoised sample is representable by a Krylov subspace, there's no need to compute the MCG. Rather, the standard CG method suffices to guarantee that the updated samples stay within the tangent space. To sum up, our DDS algorithm is as follows:

$$\boldsymbol{x}_{t-1} = \sqrt{\bar{\alpha}_{t-1}}\hat{\boldsymbol{x}}_t' + \tilde{\boldsymbol{w}}_t, \tag{23}$$

$$\hat{\boldsymbol{x}}_t' = \mathtt{CG}(\boldsymbol{A}^*\boldsymbol{A}, \boldsymbol{A}^*\boldsymbol{y}, \hat{\boldsymbol{x}}_t, M), \quad M \leq l \tag{24}$$

where $\mathtt{CG}(\cdot)$ denotes the $M$-step CG for the normal equation starting from $\hat{\boldsymbol{x}}_t$. In contrast, DDNM (Wang et al., 2023) for noiseless image restoration problems uses the following update instead of (24):

$$\hat{\boldsymbol{x}}_t' = (\boldsymbol{I} - \boldsymbol{A}^\dagger \boldsymbol{A})\hat{\boldsymbol{x}}_t + \boldsymbol{A}^\dagger \boldsymbol{y}, \tag{25}$$

where $\boldsymbol{A}^\dagger$ denotes the pseudo-inverse of $\boldsymbol{A}$. Unfortunately, (25) in DDNM does not ensure that the update signal $\hat{\boldsymbol{x}}_t'$ lies in $\mathcal{T}_t$ due to $\boldsymbol{A}^\dagger$.

Therefore, for large-scale inverse problems, we find that CG outperforms naive projections (25) by quite large margins. This is to be expected, as CG iteration enforces the update to stay on $\mathcal{T}_t$ whereas the orthogonal projections in DDNM do not guarantee this property. In practice, even when our Krylov subspace assumptions cannot be guaranteed, we empirically validate in Appendix F.1 that DDS indeed keeps $\boldsymbol{x}_t$ closest to the noisy manifold $\mathcal{M}_t$, which, in turn, shows that DDS keeps the update close to the clean manifold $\mathcal{M}$. Moreover, it is worth emphasizing that gradient-based methods (Jalal et al., 2021; Chung et al., 2022a; 2023a) often fail when choosing the "theoretically correct" step sizes of the likelihood. To fix this, several heuristics on the choice of step sizes are required (e.g. choosing the step size $\propto 1/\|\boldsymbol{y} - \boldsymbol{A}\hat{\boldsymbol{x}}_t\|_2$), which easily breaks when varying the NFE. In this regard, DDS is beneficial in that it is free from the cumbersome step-size tuning process.

| Mask Pattern | Acc. | | TV | Supervised U-Net Zbontar et al. (2018) | E2E-Varnet Sriram et al. (2020) | Jalal et al. (2100) | Score-MRI ($4000 \times 2 \times C^*$) | DPS (1000) | DDS VP (99) | DDS VP (49) | DDS VP (19) |
|---|---|---|---|---|---|---|---|---|---|---|---|
| **Uniform 1D** | $\times 4$ | PSNR [db] | 27.32 ±0.43 | 31.77±0.89 | 32.96±0.59 | 32.49±2.10 | 33.25±1.18 | 30.56±0.66 | **34.88**±0.74 | 34.61±0.32 | 32.73±2.04 |
| | | SSIM | 0.662±0.17 | 0.846±0.11 | 0.856±0.11 | 0.868±0.08 | 0.857±0.08 | 0.840±0.20 | 0.954±0.11 | **0.956**±0.08 | 0.927±0.08 |
| | $\times 8$ | PSNR [db] | 25.02±2.21 | 29.51±0.37 | 31.98±0.35 | **32.19**±2.45 | 32.01±2.30 | 30.29±0.33 | 31.62±1.88 | 30.16±1.19 | 30.33±2.35 |
| | | SSIM | 0.532±0.05 | 0.780±0.05 | 0.828±0.08 | 0.835±0.16 | 0.821±0.15 | 0.811±0.18 | 0.876±0.08 | 0.830±0.04 | **0.891**±0.16 |
| **Gaussian 1D** | $\times 4$ | PSNR [db] | 30.55±1.77 | 32.66±0.26 | 34.15±1.40 | 33.98±1.25 | 34.25±1.33 | 32.47±1.09 | 35.12±1.37 | **35.15**±0.39 | 34.63±1.95 |
| | | SSIM | 0.789±0.06 | 0.866±0.12 | 0.878±0.19 | 0.881±0.12 | 0.885±0.08 | 0.838±0.20 | **0.963**±0.15 | 0.961±0.06 | 0.957±0.09 |
| | $\times 8$ | PSNR [db] | 27.98±1.28 | 31.64±1.12 | 33.15±2.09 | 32.76±2.43 | 32.43±0.95 | 30.47±2.32 | 33.27±1.06 | **33.43**±0.75 | 32.83±1.29 |
| | | SSIM | 0.747±0.21 | 0.841±0.09 | 0.868±0.18 | 0.870±0.13 | 0.855±0.13 | 0.839±0.16 | 0.937±0.07 | **0.947**±0.15 | 0.940±0.09 |
| **Gaussian 2D** | $\times 8$ | PSNR [db] | 29.20±2.37 | 24.51±0.69 | 20.97±1.24 | 30.97±1.14 | 31.43±1.23 | 29.65±1.26 | 33.99±1.30 | **34.55**±1.69 | 32.55±1.54 |
| | | SSIM | 0.781±0.09 | 0.724±0.10 | 0.642±0.08 | 0.812±0.17 | 0.831±0.18 | 0.795±0.12 | 0.948±0.13 | **0.956**±0.15 | 0.916±0.14 |
| | $\times 15$ | PSNR [db] | 26.28±2.28 | 14.93±3.33 | 16.66±4.02 | 27.34±1.97 | **29.17**±0.98 | 26.30±1.34 | 27.86±1.67 | 25.75±1.77 | 25.66±2.03 |
| | | SSIM | 0.547±0.19 | 0.372±0.29 | 0.435±0.26 | 0.692±0.13 | 0.704±0.08 | 0.688±0.11 | **0.732**±0.10 | 0.695±0.09 | 0.693±0.08 |
| **VD Poisson disk** | $\times 8$ | PSNR [db] | 29.52±1.26 | 20.89±3.09 | 20.70±3.08 | 32.60±1.88 | 31.98±0.51 | 31.05±0.46 | 35.31±0.79 | 35.36±0.41 | **35.39**±0.57 |
| | | SSIM | 0.562±0.11 | 0.576±0.10 | 0.592±0.18 | 0.833±0.05 | 0.816±0.07 | 0.811±0.08 | 0.897±0.07 | 0.875±0.09 | **0.915**±0.11 |
| | $\times 15$ | PSNR [db] | 26.19±2.36 | 16.01±5.59 | 18.82±3.30 | 30.22±1.89 | 29.59±1.22 | 30.02±1.72 | 34.84±1.44 | **35.18**±0.97 | 34.59±1.50 |
| | | SSIM | 0.510±0.20 | 0.537±0.21 | 0.548±0.19 | 0.749±0.17 | 0.702±0.15 | 0.753±0.15 | 0.934±0.06 | 0.931±0.05 | **0.940**±0.05 |

Table 1: Quantitative metrics for noiseless parallel imaging reconstruction. Numbers in parenthesis: NFE. $^*$: expressed as $\times 2 \times C$ as the network is evaluated for real/imag part of each coil. **Bold**: best. Mean $\pm$ 1 std.

Furthermore, our CG step can be easily extended for noisy image restoration problems. Unlike the DDNM approach that relies on the singular value decomposition to handle noise, which is non-trivial to perform on forward operators in medical imaging (e.g. PI CS-MRI, CT), we can simply minimize the cost function

$$\ell(\boldsymbol{x}) = \frac{\gamma}{2} \|\boldsymbol{y} - \boldsymbol{A}\boldsymbol{x}\|_2^2 + \frac{1}{2} \|\boldsymbol{x} - \hat{\boldsymbol{x}}_t\|_2^2, \tag{26}$$

by performing CG iteration $\mathrm{CG}(\gamma\boldsymbol{A}^*\boldsymbol{A} + \boldsymbol{I}, \hat{\boldsymbol{x}}_t + \gamma\boldsymbol{A}^*\boldsymbol{y}, \hat{\boldsymbol{x}}_t, M)$ in the place of (24), where $\gamma$ is a hyper-parameter that weights the proximal regularization (Parikh & Boyd, 2014). Finally, our method can also be readily extended to accelerate DiffusionMBIR (Chung et al., 2023b) for 3D CT reconstruction by adhering to the same principles. Specifically, we implement an optimization strategy to impose the conditioning:

$$\min_{\boldsymbol{x}} \frac{1}{2} \|\boldsymbol{A}\boldsymbol{x} - \boldsymbol{y}\|_2^2 + \lambda \|\boldsymbol{D}_z\boldsymbol{x}\|_1, \tag{27}$$

where $\boldsymbol{D}_z$ is the finite difference operator that is applied to the $z$-axis that is not learned through the diffusion prior, and unlike Chung et al. (2023b), the optimization is performed in the clean manifold starting from the denoised $\hat{\boldsymbol{x}}_t$ rather than the noisy manifold starting from $\boldsymbol{x}_t$. As the additional prior is only imposed in the direction that is orthogonal to the axial slice dimension ($xy$) captured by the diffusion prior (i.e. manifold $\mathcal{M}$), (27) can be solved effectively with the alternating direction method of multipliers (ADMM) (Boyd et al., 2011) after sampling 2D diffusion slice by slice. See Appendix C for details in implementation.

## 4 EXPERIMENTS

**Problem setting.** We have the following general measurement model (see Fig. 3 for illustration of the imaging physics).

$$\boldsymbol{y} = \boldsymbol{PTs}\boldsymbol{x} =: \boldsymbol{A}\boldsymbol{x}, \ \boldsymbol{y} \in \mathbb{C}^n, \boldsymbol{A} \in \mathbb{C}^{n \times d}, \tag{28}$$

where $\boldsymbol{P}$ is the sub-sampling matrix, $\boldsymbol{T}$ is the discrete transform matrix (i.e. Fourier, Radon), and $\boldsymbol{s} = \boldsymbol{I}$ when we have a single-array measurement including CT, and $\boldsymbol{s} = [\boldsymbol{s}^{(1)}, \ldots, \boldsymbol{s}^{(c)}]$ when we have a $c-$coil parallel imaging (PI) measurement.

We conduct experiments on two distinguished applications—accelerated MRI, and 3D CT reconstruction. For the former, we follow the evaluation protocol of Chung & Ye (2022) and test our method on the fastMRI knee dataset (Zbontar et al., 2018) on diverse sub-sampling patterns. We provide comparisons against representative DIS methods Score-MRI (Chung & Ye, 2022), Jalal et al. (2021), DPS (Chung et al., 2023a). Notably, for DPS, we use the DDIM sampling strategy to show that the strength of DDS not only comes from the DDIM sampling strategy but also the use of the sampling together with the CG update steps. The optimal $\eta$ values for DPS (DDIM) are obtained through grid search. We do not compare against (Song et al., 2022) as the method cannot cover the multi-coil setting. Other than DIS, we also compare against strong supervised learning baselines: E2E-Varnet (Sriram et al., 2020), U-Net (Zbontar et al., 2018); and compressed sensing baseline: total

variation reconstruction (Block et al., 2007). For the latter, we follow Chung et al. (2023b) and test both sparse-view CT reconstruction (SV-CT), and limited angle CT reconstruction (LA-CT) on the AAPM 256×256 dataset. We compare against representative DIS methods DiffusionMBIR (Chung et al., 2023b), Song et al. (2022), MCG (Chung et al., 2022a), and DPS (Chung et al., 2023a); Supervised baselines Lahiri et al. (2022) and FBPConvNet (Jin et al., 2017); Compressed sensing baseline ADMM-TV. For all proposed methods, we employ $M = 5$, $\eta = 0.15$ for 19 NFE, $\eta = 0.5$ for 49 NFE, $\eta = 0.8$ for 99 NFE unless specified otherwise. While we only compare against a single CS baseline, it was reported in previous works that diffusion model-based solvers largely outperform the classic CS baselines (Jalal et al., 2021; Luo et al., 2023), for example, L1-wavelet (Lustig et al., 2007) and L1-ESPiRiT (Uecker et al., 2014). For PI CS-MRI experiments, we employ the rejection sampling based on a residual-based criterion to ensure stability. Further experimental details can be found in appendix G.

## 4.1 ACCELERATED MRI

**Improvement over DDNM.** Fixing the sampling strategy the same, we inspect the effect of the three different data consistency imposing strategies: Score-MRI (Chung & Ye, 2022), DDNM (Wang et al., 2023), and DDS. For DDS, we additionally search for the optimal number of CG iterations per sampling step. In Tab. 2, we see that under the low NFE regime, the score-MRI DC strategy has significantly worse performance than the proposed methods, even when using the same DDIM sampling strategy. Moreover, we see that overall, DDS outperforms DDNM by a few db in PSNR. We see that 5 CG iterations per denoising step strike a good balance. One might question the additional computational overhead of introducing the iterative CG into the already slow diffusion sampling. Nonetheless, from our experiments, we see that on average, a single CG iteration takes about 0.004 sec. Consequently, it only takes about 0.2 sec more than the analytic counterpart when using 50 NFE (Analytic: 4.51 sec vs. CG(5): 4.71 sec.).

**Improvement on VE (Chung & Ye, 2022).** Keeping the pre-trained model intact from Chung & Ye (2022), we switch from the Score-MRI sampling to Algorithm 5, and report on the reconstruction results from uniform 1D ×4 accelerated measurements in Tab. 6. Note that Score-MRI uses 2000 PC as the default setting, which amounts to 4000 NFE, reaching 33.25 PSNR. We see almost no degradation in quality down to 200 NFE, but

| | Score-MRI | DDNM | DDS (ours) | | | |
|---|---|---|---|---|---|---|
| | | | 1 | 3 | 5 | 10 |
| PSNR[db] | 26.48 | 31.36 | 31.51 | 33.78 | **34.61** | 32.48 |
| SSIM | 0.688 | 0.932 | 0.934 | 0.952 | **0.956** | 0.949 |

Table 2: Ablation study on the DC strategy. 49 NFE VP DDIM sampling strategy, uniform 1D ×4 acc. reconstruction.

the performance rapidly degrades as we move down to 100, and completely fails when we set the NFE $\leq 50$. On the other hand, by switching to the proposed solver, we are able to achieve the reconstruction quality that *better than* Score-MRI (4000 NFE) with only 100 NFE sampling. Moreover, we see that we can reduce the NFE down to 30 and still achieve decent reconstructions. This is a useful property for a reconstruction algorithm, as we can trade off reconstruction quality with speed. However, we observe several downsides of using the VE parameterization including numerical instability with large NFE, which we analyze in detail in appendix F.

**Parallel Imaging with VP parameterization (Noiseless).** We conduct thorough PI reconstruction experiments with 4 different types of sub-sampling patterns following Chung & Ye (2022). Algorithm 2 in supplementary material is used for all experiments. Quantitative results are shown in Tab. 1 (Also see Fig. 7 for qualitative results). As the proposed method is based on diffusion models, it is agnostic to the sub-sampling patterns, generalizing well to all the different sampling patterns, whereas supervised learning-based methods such as U-Net and E2E-Varnet fail dramatically on 2D subsampling patterns. Furthermore, to emphasize that the proposed method is agnostic to the imaging forward model, we show for the first time in the DIS literature that DDS is capable of reconstructing from non-cartesian MRI sub-sampling patterns that involve non-uniform Fast-Fourier Transform (NUFFT) (Fessler & Sutton, 2003). See Appendix F.2.

In Tab. 1, we see that DDS sets the new state-of-the-art in most cases even when the NFE is constrained to $< 100$. Note that this is a dramatic improvement over the previous method Chung & Ye (2022), as for parallel imaging, Score-MRI required $120k(C = 15)$ NFE for the reconstruction of a single image. Contrarily, DDS is able to *outperform* score-MRI with 49 NFE, and performs

| Method | 8-view | | | | | | 4-view | | | | | | 2-view | | | | | |
|---|---|---|---|---|---|---|---|---|---|---|---|---|---|---|---|---|---|---|
| | Axial* | | Coronal | | Sagittal | | Axial* | | Coronal | | Sagittal | | Axial* | | Coronal | | Sagittal | |
| | PSNR↑ | SSIM↑ | PSNR↑ | SSIM↑ | PSNR↑ | SSIM↑ | PSNR↑ | SSIM↑ | PSNR↑ | SSIM↑ | PSNR↑ | SSIM↑ | PSNR↑ | SSIM↑ | PSNR↑ | SSIM↑ | PSNR↑ | SSIM↑ |
| DDS VP (19) | 32.31 | 0.904 | **35.82** | **0.975** | **33.03** | 0.931 | 30.59 | 0.906 | 30.38 | 0.947 | 27.90 | 0.903 | 25.43 | 0.844 | 24.38 | 0.862 | 22.10 | 0.769 |
| DDS VP (49) | **33.86** | 0.930 | 35.39 | 0.974 | 32.97 | **0.937** | **31.48** | **0.918** | **30.81** | 0.949 | 28.43 | 0.900 | **25.85** | **0.857** | **24.60** | **0.871** | **22.69** | **0.791** |
| DDS VE (99) | 33.43 | 0.932 | 34.35 | 0.972 | 32.01 | 0.935 | 31.23 | 0.915 | 30.62 | **0.958** | **28.52** | **0.914** | 25.09 | 0.833 | 24.15 | 0.855 | 22.26 | 0.785 |
| DiffusionMBIR (Chung et al., 2023b) (4000) | 33.49 | **0.942** | 35.18 | 0.967 | 32.18 | 0.910 | 30.52 | 0.914 | 30.09 | 0.938 | 27.89 | 0.871 | 24.11 | 0.810 | 23.15 | 0.841 | 21.72 | 0.766 |
| DPS (Chung et al., 2023a) (1000) | 27.86 | 0.858 | 27.07 | 0.860 | 23.66 | 0.744 | 26.96 | 0.842 | 26.09 | 0.817 | 22.70 | 0.737 | 22.16 | 0.773 | 21.56 | 0.784 | 19.34 | 0.698 |
| Score-Med (Song et al., 2022) (4000) | 29.10 | 0.882 | 27.93 | 0.875 | 24.23 | 0.759 | 28.20 | 0.867 | 27.48 | 0.889 | 25.08 | 0.783 | 24.07 | 0.808 | 23.70 | 0.822 | 20.95 | 0.720 |
| MCG (Chung et al., 2022a) (4000) | 28.61 | 0.873 | 28.05 | 0.884 | 24.45 | 0.765 | 27.33 | 0.855 | 26.52 | 0.863 | 23.04 | 0.745 | 24.69 | 0.821 | 23.52 | 0.806 | 20.71 | 0.685 |
| Lahiri et al. (2022) | 21.38 | 0.711 | 23.89 | 0.769 | 20.81 | 0.716 | 20.37 | 0.652 | 21.41 | 0.721 | 18.40 | 0.665 | 19.74 | 0.631 | 19.92 | 0.720 | 17.34 | 0.650 |
| FBPConvNet (Jin et al., 2017) | 16.57 | 0.553 | 19.12 | 0.774 | 18.11 | 0.714 | 16.45 | 0.529 | 19.47 | 0.713 | 15.48 | 0.610 | 16.31 | 0.521 | 17.05 | 0.521 | 11.07 | 0.483 |
| ADMM-TV | 16.79 | 0.645 | 18.95 | 0.772 | 17.27 | 0.716 | 13.59 | 0.618 | 15.23 | 0.682 | 14.60 | 0.638 | 10.28 | 0.409 | 13.77 | 0.616 | 11.49 | 0.553 |

Table 4: Quantitative evaluation of SV-CT on the AAPM 256×256 test set (mean values; std values in Tab. 8). (Numbers in parenthesis): NFE, **Bold**: best.

*on par* with score-MRI with 19 NFE. Even disregarding the additional $\times 2C$ more NFEs required for score-MRI to account for the multi-coil complex valued acquisition, the proposed method still achieves $\times 80 \sim \times 200$ acceleration. We note that on average, our method takes about 4.7 seconds for 49 NFE, and about 2.25 seconds for 19 NFE on a single commodity GPU (RTX 3090).

**Noisy multi-coil MRI reconstruction.** One of the most intriguing properties of the proposed DDS is the ease of handling measurement noise without careful computation of singular value decomposition (SVD), which is non-trivial to perform for our tasks. With (26), we can solve it with CG, arriving at Algorithm 3 in supplementary material. For comparison, methods that try to cope with measurement noise via SVD in the diffusion model context (Kawar et al., 2022; Wang et al., 2023) are not applicable and cannot be compared. One work that does not require computation of SVD and hence is applicable is DPS (Chung et al., 2023a) relying on backpropagation. To test the efficacy of DDS on noisy inverse problems, we add a rather heavy complex Gaussian noise ($\sigma = 0.05$) to the $k$-space multi-coil measurements and reconstruct with Algorithm 3 by setting $\gamma = 0.95$ found through grid search. In Tab. 3, we see that DDS far outperforms DPS (Chung et al., 2023a) with 1000 NFE by a large margin, while being about $\times 40$ faster as DPS requires the heavy computation of backpropagation.

| Mask Pattern | Acc. | | TV | DPS (1000) | DDS VP (49) |
|---|---|---|---|---|---|
| Uniform 1D | ×4 | PSNR [db] | 24.19 | 24.40 | **29.47** |
| | | SSIM | 0.687 | 0.656 | **0.866** |
| | ×8 | PSNR [db] | 23.02 | 24.60 | **26.77** |
| | | SSIM | 0.638 | 0.666 | **0.827** |
| VD Poisson disk | ×8 | PSNR [db] | 23.07 | 23.48 | **30.95** |
| | | SSIM | 0.609 | 0.592 | **0.890** |
| | ×15 | PSNR [db] | 20.92 | 23.57 | **29.36** |
| | | SSIM | 0.554 | 0.622 | **0.853** |

Table 3: Quantitative metrics for **noisy** parallel imaging reconstruction. Numbers in parenthesis: NFE.

## 4.2 3D CT RECONSTRUCTION

**Sparse-view CT.** Similar to the accelerated MRI experiments, we aim to both 1) improve the original VE model of Chung et al. (2023b), and 2) train a new VP model better suitable for DDS. Inspecting Tab. 4, we see that by using Algorithm 6 in supplementary material, we are able to reduce the NFE to 100 and still achieve results that are on par with DiffusionMBIR with 4000 NFE. However, we observe similar instabilities with the VE parameterization. Additionally, we find it crucial to initialize the optimization process with CG and later switch to ADMM-TV using a CG solver for proper convergence (see appendix E.2 for discussion). Switching to the VP parameterization and using Algorithm 4, we now see that DDS achieves the new state-of-the-art with $\leq 49$ NFE. Notably, this decreases the sampling time to $\sim 25$ min for 49 NFE, and $\sim 10$ min wall-clock time for 19 NFE on a single RTX 3090 GPU, compared to the painful 2 days for DiffusionMBIR. In Tab. 7, we see similar improvements that were seen from SV-CT, where DDS significantly outperforms DiffusionMBIR while being several orders of magnitude faster.

## 5 CONCLUSION

In this work, we present Decomposed Diffusion Sampling (DDS), a general DIS for challenging real-world medical imaging inverse problems. Leveraging the geometric view of diffusion models and the property of the CG solvers on the tangent space, we show that performing numerical optimization schemes on the denoised representation is superior to the previous methods of imposing DC. Further, we devise a fast sampler based on DDIM that works well for both VE/VP settings. With extensive experiments on multi-coil MRI reconstruction and 3D CT reconstruction, we show that DDS achieves superior quality while being $\geq \times 80$ faster than the previous DIS.

**Ethics Statement**   We recognize the profound potential of our approach to revolutionize diagnostic procedures, enhance patient care, and reduce the need for invasive techniques. However, we are also acutely aware of the ethical considerations surrounding patient data privacy and the potential for misinterpretation of generated images. All medical data used in our experiments were publicly available and fully anonymized, ensuring the utmost respect for patient confidentiality. We advocate for rigorous validation and clinical collaboration before any real-world application of our findings, to ensure both the safety and efficacy of our proposed methods in a medical setting.

**Reproducibility Statement**   For every different application and different circumstances (noise-less/noisy, VE/VP, 2D/3D), we provide tailored algorithms(See Appendix. C,E) to ensure maximum reproducibility. All the hyper-parameters used in the algorithms are detailed in Section 4 and Appendix G. Code is open-sourced at `https://github.com/HJ-harry/DDS`.

ACKNOWLEDGMENTS

This research was supported by the National Research Foundation of Korea(NRF)(RS-2023-00262527), Field-oriented Technology Development Project for Customs Administration funded by the Korean government (the Ministry of Science & ICT and the Korea Customs Service) through the National Research Foundation (NRF) of Korea under Grant NRF2021M3I1A1097910 & NRF-2021M3I1A1097938, Korea Medical Device Development Fund grant funded by the Korea government (the Ministry of Science and ICT, the Ministry of Trade, Industry, and Energy, the Ministry of Health & Welfare, the Ministry of Food and Drug Safety) (Project Number: 1711137899, KMDF_PR_20200901_0015), and Culture, Sports, and Tourism R&D Program through the Korea Creative Content Agency grant funded by the Ministry of Culture, Sports and Tourism in 2023.

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

# A PRELIMINARIES

## A.1 DIFFUSION MODELS

Let us define a random variable $\boldsymbol{x}_0 \sim p(\boldsymbol{x}_0) = p_{\text{data}}(\boldsymbol{x}_0)$, where $p_{\text{data}}$ denotes the data distribution. In diffusion models, we construct a continuous Gaussian perturbation kernel $p(\boldsymbol{x}_t|\boldsymbol{x}_0) = \mathcal{N}(\boldsymbol{x}_t; s_t\boldsymbol{x}_0, s_t^2\sigma_t^2\boldsymbol{I})$ with $t \in [0, 1]$, which smooths out the distribution. As $t \to 1$, the marginal distribution $p_t(\boldsymbol{x}_t)$ is smoothed such that it approximates the Gaussian distribution, which becomes our reference distribution to sample from. Using the reparametrization trick, one can directly sample

$$\boldsymbol{x}_t = s_t\boldsymbol{x}_0 + s_t\sigma_t\boldsymbol{z}, \quad \boldsymbol{z} \sim \mathcal{N}(0, \boldsymbol{I}). \tag{29}$$

Diffusion models aim to revert the data noising process. Remarkably, it was shown that the data noising process and the denoising process can both be represented as a stochastic differential equation (SDE), governed by the score function $\nabla_{\boldsymbol{x}_t} \log p(\boldsymbol{x}_t)$ (Song et al., 2021b; Karras et al., 2022). Namely, the forward/reverse diffusion SDE can be succinctly represented as (assuming $s_t = 1$ for simplicity,

$$d\boldsymbol{x}_\pm = -\dot{\sigma}_t\sigma_t\nabla_{\boldsymbol{x}_t} \log p(\boldsymbol{x}_t)\,dt \pm \dot{\sigma}_t\sigma_t\nabla_{\boldsymbol{x}_t} \log p(\boldsymbol{x}_t)\,dt + \sqrt{\dot{\sigma}_t\sigma_t}d\boldsymbol{w}_t, \tag{30}$$

where $\boldsymbol{w}_t$ is the standard Wiener process. Here, the $+$ sign denotes the forward process, where (30) collapses to a Brownian motion. With the $-$ sign, the process runs backward, and we see that the score function $\nabla_{\boldsymbol{x}_t} \log p(\boldsymbol{x}_t)$ governs the reverse SDE. In other words, in order to run reverse diffusion sampling (i.e. generative modeling), we need access to the score function of the data distribution.

The procedure called score matching, where one tries to train a parametrized model $\boldsymbol{s}_\theta$ to approximate $\nabla_{\boldsymbol{x}_t} \log p(\boldsymbol{x}_t)$ can be done through score matching (Hyvärinen & Dayan, 2005). As explicit and implicit score matching methods are costly to perform, the most widely used training method in the modern sense is the so-called denoising score matching (DSM) (Vincent, 2011)

$$\min_\theta \mathbb{E}_{\boldsymbol{x}_t,\boldsymbol{x}_0,\epsilon} \left[ \|\boldsymbol{s}_\theta^{(t)}(\boldsymbol{x}_t) - \nabla_{\boldsymbol{x}_t} \log p(\boldsymbol{x}_t|\boldsymbol{x}_0)\|_2^2 \right], \tag{31}$$

which is easy to train as our perturbation kernel is Gaussian. Once $\boldsymbol{s}_{\theta^*}$ is trained, we can use it as a plug-in approximation of the score function to plug into (30).

The score function has close relation to the posterior mean $\mathbb{E}[\boldsymbol{x}_0|\boldsymbol{x}_t]$, which can be formally linked through Tweedie's formula

**Lemma 1** (Tweedie's formula). *Given a Gaussian perturbation kernel* $p(\boldsymbol{x}_t|\boldsymbol{x}_0) = \mathcal{N}(\boldsymbol{x}_t; s_t\boldsymbol{x}_0, \sigma_t^2\boldsymbol{I})$, *the posterior mean is given by*

$$\mathbb{E}[\boldsymbol{x}_0|\boldsymbol{x}_t] = \frac{1}{s_t}(\boldsymbol{x}_t + \sigma_t^2\nabla_{\boldsymbol{x}_t} \log p(\boldsymbol{x}_t)) \tag{32}$$

*Proof.*

$$\nabla_{\boldsymbol{x}_t} \log p(\boldsymbol{x}_t) = \frac{\nabla_{\boldsymbol{x}_t} p(\boldsymbol{x}_t)}{p(\boldsymbol{x}_t)} \tag{33}$$

$$= \frac{1}{p(\boldsymbol{x}_t)}\nabla_{\boldsymbol{x}_t} \int p(\boldsymbol{x}_t|\boldsymbol{x}_0)p(\boldsymbol{x}_0)\,d\boldsymbol{x}_0 \tag{34}$$

$$= \frac{1}{p(\boldsymbol{x}_t)} \int \nabla_{\boldsymbol{x}_t} p(\boldsymbol{x}_t|\boldsymbol{x}_0)p(\boldsymbol{x}_0)\,d\boldsymbol{x}_0 \tag{35}$$

$$= \frac{1}{p(\boldsymbol{x}_t)} \int p(\boldsymbol{x}_t|\boldsymbol{x}_0)\nabla_{\boldsymbol{x}_t} \log p(\boldsymbol{x}_t|\boldsymbol{x}_0)p(\boldsymbol{x}_0)\,d\boldsymbol{x}_0 \tag{36}$$

$$= \int p(\boldsymbol{x}_0|\boldsymbol{x}_t)\nabla_{\boldsymbol{x}_t} \log p(\boldsymbol{x}_t|\boldsymbol{x}_0)\,d\boldsymbol{x}_0 \tag{37}$$

$$= \int p(\boldsymbol{x}_0|\boldsymbol{x}_t)\frac{s_t\boldsymbol{x}_0 - \boldsymbol{x}_t}{\sigma_t^2}\,d\boldsymbol{x}_0 \tag{38}$$

$$= \frac{s_t\mathbb{E}[\boldsymbol{x}_0|\boldsymbol{x}_t] - \boldsymbol{x}_t}{\sigma_t^2}. \tag{39}$$

$\square$

In other words, having access to the score function is equivalent to having access to the posterior mean through time $t$, which we extensively leverage in this work.

### A.2 KRYLOV SUBSPACE METHODS

Consider the problem $\boldsymbol{y} = \boldsymbol{A}\boldsymbol{x}$, where $\boldsymbol{x}, \boldsymbol{y} \in \mathbb{R}^N$, and we have a square matrix $\boldsymbol{A} \in \mathbb{R}^{N \times N}$. We define a fixed point iteration

$$\boldsymbol{x} = \hat{\boldsymbol{B}}\boldsymbol{x} + \hat{\boldsymbol{y}}, \quad \hat{\boldsymbol{B}} := \boldsymbol{I} - \boldsymbol{D}^{-1}\boldsymbol{A}, \hat{\boldsymbol{y}} := \boldsymbol{D}^{-1}\boldsymbol{y}, \tag{40}$$

where $\boldsymbol{D}$ is some diagonal matrix. One can show that the iteration in (40) converges to the solution of $\boldsymbol{x}^* = \boldsymbol{A}^{-1}\boldsymbol{y}$ if and only if $\rho(\hat{\boldsymbol{B}}) < 1$, where $\rho(\cdot)$ denotes the spectral norm of the matrix. More specifically, we further have that

$$\limsup_{n \to \infty} \|\boldsymbol{x}_n - \boldsymbol{x}^*\|^{1/n} \leq \rho(\hat{\boldsymbol{B}}). \tag{41}$$

Here, unless we know the solution $\boldsymbol{x}^*$, we cannot compute the error vector

$$\boldsymbol{d}_n := \boldsymbol{x}_n - \boldsymbol{x}^*. \tag{42}$$

Instead, however, what we have access to is the *residual*

$$\boldsymbol{b}_n = \boldsymbol{y} - \boldsymbol{A}\boldsymbol{x}_n = -\boldsymbol{A}\boldsymbol{d}_n. \tag{43}$$

For simplicity, let $\boldsymbol{D} = \boldsymbol{I}$ and thus $\hat{\boldsymbol{B}} = \boldsymbol{I} - \boldsymbol{A}$. The residual now becomes

$$\boldsymbol{b}_n = \boldsymbol{y} - \boldsymbol{A}\boldsymbol{x}_n = \hat{\boldsymbol{B}}\boldsymbol{x}_n + \boldsymbol{y} - \boldsymbol{x}_n = \boldsymbol{x}_{n+1} - \boldsymbol{x}_n. \tag{44}$$

Then, with (44), the Jacobi iteration reads

$$\boldsymbol{x}_{n+1} := \boldsymbol{x}_n + \boldsymbol{b}_n \tag{45}$$

$$\boldsymbol{b}_{n+1} := \boldsymbol{b}_n - \boldsymbol{A}\boldsymbol{b}_n = \hat{\boldsymbol{B}}\boldsymbol{b}_n. \tag{46}$$

This also implies the following

$$\boldsymbol{b}_n \in \mathrm{Span}(\boldsymbol{b}, \boldsymbol{A}\boldsymbol{b}, \cdots, \boldsymbol{A}^n\boldsymbol{b}) \tag{47}$$

$$\boldsymbol{x}_n \in \boldsymbol{x}_0 + \mathrm{Span}(\boldsymbol{b}, \boldsymbol{A}\boldsymbol{b}, \cdots, \boldsymbol{A}^{n-1}\boldsymbol{b}), \tag{48}$$

with the latter obtained by shifting the subspace of $\boldsymbol{b}_{n-1}$. The Krylov subspace is exactly defined by this span, i.e. $\mathcal{K}_n(\boldsymbol{A}) := \mathrm{Span}(\boldsymbol{b}, \boldsymbol{A}\boldsymbol{b}, \cdots, \boldsymbol{A}^n\boldsymbol{b})$.

In Krylov subspace methods, the idea is to generate a sequence of approximate solutions $\boldsymbol{x}_n \in \boldsymbol{x}_0 + \mathcal{K}_n(\boldsymbol{A})$ of $\boldsymbol{A}\boldsymbol{x} = \boldsymbol{y}$, so that the sequence of the residuals $\mathrm{b}_n \in \mathcal{K}_{n+1}(\boldsymbol{A})$ converges to the zero vector.

## B PROOFS

**Lemma 2** (Total noise). *Assuming optimality of $\epsilon_{\theta^*}^{(t)}$, the total noise $\tilde{\boldsymbol{w}}_t$ in (13) can be represented by*

$$\tilde{\boldsymbol{w}}_t = \sqrt{1 - \bar{\alpha}_{t-1}}\tilde{\boldsymbol{\epsilon}} \tag{49}$$

*for some $\tilde{\boldsymbol{\epsilon}} \sim \mathcal{N}(0, \boldsymbol{I})$. In other words, (23) is equivalently represented by $\boldsymbol{x}_{t-1} = \sqrt{\bar{\alpha}_{t-1}}\hat{\boldsymbol{x}}_t + \sqrt{1 - \bar{\alpha}_{t-1}}\tilde{\boldsymbol{\epsilon}}$ for some $\tilde{\boldsymbol{\epsilon}} \sim \mathcal{N}(0, \boldsymbol{I})$.*

*Proof.* Given the independence of the estimated $\hat{\boldsymbol{\epsilon}}_t$ and the stochastic $\boldsymbol{\epsilon} \sim \mathcal{N}(0, \boldsymbol{I})$ along with the Gaussianity, the noise variance in (15) is given as $1 - \bar{\alpha}_{t-1} - \eta^2\tilde{\beta}_t^2 + \eta^2\tilde{\beta}_t^2 = 1 - \bar{\alpha}_{t-1}$, recovering a sample from $q(\boldsymbol{x}_{t-1}|\boldsymbol{x}_0)$. $\square$

**Proposition 1** (Manifold Constrained Gradient). *Suppose the clean data manifold $\mathcal{M}$ is represented as an affine subspace and assumes the uniform distribution on $\mathcal{M}$. Then,*

$$\frac{\partial \hat{\boldsymbol{x}}_t}{\partial \boldsymbol{x}_t} = \frac{1}{\sqrt{\bar{\alpha}_t}}\mathcal{P}_{\mathcal{M}} \tag{19}$$

$$\hat{\boldsymbol{x}}_t - \gamma_t \nabla_{\boldsymbol{x}_t}\ell(\hat{\boldsymbol{x}}_t) = \mathcal{P}_{\mathcal{M}}(\hat{\boldsymbol{x}}_t - \zeta_t\nabla_{\hat{\boldsymbol{x}}_t}\ell(\hat{\boldsymbol{x}}_t)) \tag{20}$$

*for some $\zeta_t > 0$, where $\mathcal{P}_{\mathcal{M}}$ denotes the orthogonal projection to $\mathcal{M}$.*

*Proof.* First, we provide a proof for (19). Thanks to the forward sampling $\boldsymbol{x}_{t-1} = \sqrt{\bar{\alpha}_{t-1}}\boldsymbol{x}_0 + \sqrt{1 - \bar{\alpha}_{t-1}}\tilde{\boldsymbol{\epsilon}}$, we can obtain the closed-form expression of the likelihood:

$$p(\boldsymbol{x}_t|\boldsymbol{x}_0) = \frac{1}{(2\pi(1-\bar{\alpha}(t)))^{d/2}} \exp\left(-\frac{\|\boldsymbol{x}_t - \sqrt{\bar{\alpha}(t)}\boldsymbol{x}_0\|^2}{2(1-\bar{\alpha}(t))}\right), \tag{50}$$

which is a Gaussian distribution. Using the Bayes' theorem, we have

$$p(\boldsymbol{x}_t) = \int p(\boldsymbol{x}_t|\boldsymbol{x}_0)p(\boldsymbol{x}_0)d\boldsymbol{x}_0. \tag{51}$$

According to the assumption, $p(\boldsymbol{x}_0)$ is a uniform distribution on the subspace $\mathcal{M}$. To incorporate this in (51), we first compute $p(\boldsymbol{x}_t)$ by modeling $p(\boldsymbol{x}_0)$ as the zero-mean Gaussian distribution with the isotropic variance $\sigma^2$ and then take the limit $\sigma \to \infty$. More specifically, we have

$$p(\boldsymbol{x}_0) = \frac{1}{(2\pi\sigma^2)^{l/2}} \exp\left(-\frac{\|\mathcal{P}_{\mathcal{M}}\boldsymbol{x}_0\|^2}{2\sigma^2}\right), \tag{52}$$

where we use $\mathcal{P}_{\mathcal{M}}\boldsymbol{x}_0 = \boldsymbol{x}_0$ as $\boldsymbol{x}_0 \in \mathcal{M}$. Therefore, we have

$$p(\boldsymbol{x}_t, \boldsymbol{x}_0) = p(\boldsymbol{x}_t|\boldsymbol{x}_0)p(\boldsymbol{x}_0)$$
$$= \frac{1}{(2\pi(1-\bar{\alpha}(t)))^{d/2}} \frac{1}{(2\pi\sigma^2)^{l/2}} \exp\left(-d(\boldsymbol{x}_t, \boldsymbol{x}_0)\right)$$

where

$$d(\boldsymbol{x}_t, \boldsymbol{x}_0)$$
$$= \frac{\|\mathcal{P}_{\mathcal{M}}^{\perp}\boldsymbol{x}_t\|^2}{2(1-\bar{\alpha}(t))} + \frac{\|\mathcal{P}_{\mathcal{M}}\boldsymbol{x}_0\|^2}{2\sigma^2} + \frac{\|\mathcal{P}_{\mathcal{M}}\boldsymbol{x}_t - \sqrt{\bar{\alpha}(t)}\mathcal{P}_{\mathcal{M}}\boldsymbol{x}_0\|^2}{2(1-\bar{\alpha}(t))}$$
$$= \frac{\|\mathcal{P}_{\mathcal{M}}\boldsymbol{x}_0 - \boldsymbol{\mu}_t\|^2}{2s(t)} + \frac{\|\mathcal{P}_{\mathcal{M}}^{\perp}\boldsymbol{x}_t\|^2 + c(t)\|\mathcal{P}_{\mathcal{M}}\boldsymbol{x}_t\|^2}{2(1-\bar{\alpha}(t))}$$

and

$$s(t) = \left(\frac{1}{\sigma^2} + \frac{\bar{\alpha}(t)}{1-\bar{\alpha}(t)}\right)^{-1} \tag{53}$$

$$\boldsymbol{\mu}_t = \frac{\frac{\sqrt{\bar{\alpha}(t)}}{1-\bar{\alpha}(t)}}{\frac{1}{\sigma^2} + \frac{\bar{\alpha}(t)}{1-\bar{\alpha}(t)}}\boldsymbol{x}_t \tag{54}$$

$$c(t) = \frac{1}{1 + \frac{\bar{\alpha}(t)}{1-\bar{\alpha}(t)}\sigma^2} \tag{55}$$

Therefore, after integrating out with respect to $\boldsymbol{x}_0$, we have

$$\log p(\boldsymbol{x}_t) = -\frac{\|\mathcal{P}_{\mathcal{M}}^{\perp}\boldsymbol{x}_t\|^2 + c(t)\|\mathcal{P}_{\mathcal{M}}\boldsymbol{x}_t\|^2}{2(1-\bar{\alpha}(t))} + \text{const.}$$

leading to

$$\nabla_{\boldsymbol{x}_t} \log p(\boldsymbol{x}_t) = -\frac{1}{1-\bar{\alpha}(t)}\mathcal{P}_{\mathcal{M}}^{\perp}\boldsymbol{x}_t - \frac{c(t)}{1-\bar{\alpha}(t)}\mathcal{P}_{\mathcal{M}}\boldsymbol{x}_t$$

Furthermore, using (55), for the uniform distribution we have $\lim_{\sigma \to \infty} c(t) = 0$. Therefore,

$$\lim_{\sigma \to \infty} \nabla_{\boldsymbol{x}_t} \log p(\boldsymbol{x}_t) = -\frac{1}{1-\bar{\alpha}(t)}\mathcal{P}_{\mathcal{M}}^{\perp}\boldsymbol{x}_t$$

Now, using Tweedie's denoising formula in (14), we have

$$\hat{\boldsymbol{x}}_t = \frac{1}{\sqrt{\bar{\alpha}_t}}(\boldsymbol{x}_t - \sqrt{1-\bar{\alpha}_t}\boldsymbol{\epsilon}_{\theta^*}^{(t)}(\boldsymbol{x}_t)) \tag{56}$$

$$= \frac{1}{\sqrt{\bar{\alpha}_t}}(\boldsymbol{x}_t + (1-\bar{\alpha}_t)\nabla_{\boldsymbol{x}_t} \log p(\boldsymbol{x}_t)) \tag{57}$$

where we use

$$s_{\theta^*}^{(t)}(\boldsymbol{x}_t) = \nabla_{\boldsymbol{x}_t} \log p(\boldsymbol{x}_t) = -\boldsymbol{\epsilon}_{\theta^*}^{(t)}(\boldsymbol{x}_t)/\sqrt{1-\bar{\alpha}_t}$$

Accordingly, we have

$$\hat{\boldsymbol{x}}_t = \frac{1}{\sqrt{\bar{\alpha}_t}}(\boldsymbol{x}_t - \mathcal{P}_{\mathcal{M}}^{\perp}\boldsymbol{x}_t) = \frac{1}{\sqrt{\bar{\alpha}_t}}\mathcal{P}_{\mathcal{M}}\boldsymbol{x}_t \tag{58}$$

Therefore, we have

$$\frac{\partial \hat{\boldsymbol{x}}_t}{\partial \boldsymbol{x}_t} = \frac{1}{\sqrt{\bar{\alpha}_t}}\mathcal{P}_{\mathcal{M}} \ . \tag{59}$$

Second, we provide a proof for (20). Since $\hat{\boldsymbol{x}}_t \in \mathcal{M}$, we have $\hat{\boldsymbol{x}}_t = \mathcal{P}_{\mathcal{M}}\hat{\boldsymbol{x}}_t$. Thus, using (59), we have

$$
\begin{aligned}
\hat{\boldsymbol{x}}_t - \gamma_t \nabla_{\boldsymbol{x}_t} \ell(\hat{\boldsymbol{x}}_t) &= \hat{\boldsymbol{x}}_t - \gamma_t \frac{\partial \hat{\boldsymbol{x}}_t}{\partial \boldsymbol{x}_t} \nabla_{\hat{\boldsymbol{x}}_t} \ell(\hat{\boldsymbol{x}}_t) \\
&= \mathcal{P}_{\mathcal{M}}\hat{\boldsymbol{x}}_t - \frac{\gamma_t}{\sqrt{\bar{\alpha}_t}} \mathcal{P}_{\mathcal{M}} \nabla_{\hat{\boldsymbol{x}}_t} \ell(\hat{\boldsymbol{x}}_t) \\
&= \mathcal{P}_{\mathcal{M}}\left(\hat{\boldsymbol{x}}_t - \zeta_t \nabla_{\hat{\boldsymbol{x}}_t} \ell(\hat{\boldsymbol{x}}_t)\right)
\end{aligned}
$$

where $\zeta_t := \gamma_t/\sqrt{\bar{\alpha}_t}$. Q.E.D. $\qquad\square$

For score functions trained in the context of DDPM (VP-SDE), DDIM sampling of (13) can directly be used. However, for VE-SDE that was not developed under the variational framework, it is unclear how to construct a sampler equivalent of DDIM for VP-SDE. As one of our goals is to enable the direct adoption of pre-trained diffusion models regardless of the framework, our goal is to devise a fast sampler tailored for VE-SDE. Now, the idea of the decomposition of DDIM steps can be easily adopted to address this issue.

First, observe that the forward conditional density for VE-SDE can be written as $q(\boldsymbol{x}_t|\boldsymbol{x}_0) = \mathcal{N}(\boldsymbol{x}_t|\boldsymbol{x}_0, \sigma_t^2\boldsymbol{I})$, where $\sigma_t$ is taken to be a geometric series following Song et al. (2021b). This directly leads to the following result:

**Proposition 2** (VE Decomposition). *The following update rule recovers a sample from the marginal* $q(\boldsymbol{x}_{t-1}|\boldsymbol{x}_0) \ \forall \eta \in [0,1]$.

$$\boldsymbol{x}_{t-1} = \hat{\boldsymbol{x}}_t + \tilde{\boldsymbol{w}}_t \tag{60}$$

*where*

$$\hat{\boldsymbol{x}}_t := \boldsymbol{x}_t + \sigma_t^2 \hat{\boldsymbol{s}}_t \tag{61}$$

$$\tilde{\boldsymbol{w}}_t = -\sigma_{t-1}\sigma_t\sqrt{1-\tilde{\beta}^2\eta^2}\hat{\boldsymbol{s}}_t + \sigma_{t-1}\eta\tilde{\beta}_t\boldsymbol{\epsilon} \tag{62}$$

*Proof.* From the equivalent parameterization as in (14), we have the following relations in VE-SDE

$$\hat{\boldsymbol{s}}_t = -\frac{\boldsymbol{x}_t - \hat{\boldsymbol{x}}_t}{\sigma_t^2} \tag{63}$$

$$\hat{\boldsymbol{\epsilon}}_t = \frac{\boldsymbol{x}_t - \hat{\boldsymbol{x}}_t}{\sigma_t} = -\sigma_t \hat{\boldsymbol{s}}_t \tag{64}$$

$$\hat{\boldsymbol{x}}_t = \boldsymbol{x}_t + \sigma_t^2 \hat{\boldsymbol{s}}_t. \tag{65}$$

Plugging in, we have

$$\boldsymbol{x}_{t-1} = \hat{\boldsymbol{x}}_t + \sigma_{t-1}\sqrt{1-\tilde{\beta}^2\eta^2}\hat{\boldsymbol{\epsilon}}_t + \sigma_{t-1}\eta\tilde{\beta}\boldsymbol{\epsilon}. \tag{66}$$

Since the variance can be now computed as

$$\left(\sigma_{t-1}\sqrt{1-\tilde{\beta}^2\eta^2}\right)^2 + \left(\sigma_{t-1}\eta\tilde{\beta}\right)^2 = \sigma_{t-1}^2, \tag{67}$$

$\boldsymbol{x}_{t-1} \sim q(\boldsymbol{x}_{t-1}; \hat{\boldsymbol{x}}_t, \sigma_{t-1}^2\boldsymbol{I}) = q(\boldsymbol{x}_{t-1}|\boldsymbol{x}_0)$ by the assumption. $\qquad\square$

Again, $\hat{\boldsymbol{x}}_t$ arises from Tweedie's formula. With $\eta = 1$ and $\tilde{\beta} = 1 - \sigma_{t-1}^2/\sigma_t^2$, we can recover the original VE-SDE, whereas $\eta = 0$ leads to the deterministic variation of VE-SDE, which we call VE-DDIM. We can now use the usual VP-DDIM (13) or VE-DDIM (60) depending on the training strategy of the pre-trained score function.

Here, we present our main geometric observations in the VE context, which are analogous to Proposition 1 and Lemma 2. Their proofs are straightforward corollaries, and hence, are omitted.

**Proposition 3** (VE-DDIM Decomposition). *Under the same assumption of Proposition 1, we have*

$$\frac{\partial \hat{\boldsymbol{x}}_t}{\partial \boldsymbol{x}_t} = \mathcal{P}_{\mathcal{M}} \tag{68}$$

$$\hat{\boldsymbol{x}}_t - \gamma_t \nabla_{\boldsymbol{x}_t} \ell(\hat{\boldsymbol{x}}_t) = \mathcal{P}_{\mathcal{M}} (\hat{\boldsymbol{x}}_t - \gamma_t \nabla_{\hat{\boldsymbol{x}}_t} \ell(\hat{\boldsymbol{x}}_t)) \tag{69}$$

*where $\mathcal{P}_{\mathcal{M}}$ denotes the orthogonal projection to the subspace $\mathcal{M}$.*

Accordingly, our DDS algorithm for VE-DDIM is as follows:

$$\boldsymbol{x}_{t-1} = \hat{\boldsymbol{x}}_t' + \tilde{\boldsymbol{w}}_t, \tag{70}$$

$$\hat{\boldsymbol{x}}_t' = \text{CG}(\boldsymbol{A}^*\boldsymbol{A}, \boldsymbol{A}^*\boldsymbol{y}, \hat{\boldsymbol{x}}_t, M), \quad M \leq l \tag{71}$$

where $\text{CG}(\cdot)$ denotes the $M$-step CG for the normal equation starting from $\hat{\boldsymbol{x}}_t$.

## C  ALGORITHMS

In the following tables, we list all the DDS algorithms used throughout the manuscript. For simplicity, we define $\text{CG}(\boldsymbol{A}, \boldsymbol{y}, \boldsymbol{x}, M)$ to be running $M$ steps of conjugate gradient steps with initialization $\boldsymbol{x}$. For completeness, we include a pseudo-code of the CG method in Algorithm. 1 that is used throughout the work.

---

**Algorithm 1** Conjugate Gradient (CG)

---

**Require:** $\boldsymbol{A}, \boldsymbol{y}, \boldsymbol{x}_0, M$
1: $\boldsymbol{r}_0 \leftarrow \boldsymbol{b} - \boldsymbol{A}\boldsymbol{x}_0$
2: $\boldsymbol{p}_0 \leftarrow \mathsf{b}_0$
3: **for** $i = 0 : K - 1$ **do**
4: $\quad \alpha_k \leftarrow \frac{\boldsymbol{r}_k^\top \boldsymbol{r}}{\boldsymbol{p}_k^\top \boldsymbol{A}\boldsymbol{p}_k}$
5: $\quad \boldsymbol{x}_{k+1} \leftarrow \boldsymbol{x}_k + \alpha_k \boldsymbol{p}_k$
6: $\quad \boldsymbol{r}_{k+1} \leftarrow \boldsymbol{r}_k - \alpha_k \boldsymbol{A}\boldsymbol{p}_k$
7: $\quad \beta_k \leftarrow \frac{\boldsymbol{r}_k^\top \boldsymbol{r}}{\boldsymbol{r}_k^\top \boldsymbol{r}_k}$
8: $\quad \boldsymbol{p}_{k+1} \leftarrow \mathsf{b}_{k+1} + \beta_k \boldsymbol{p}_k$
9: **end for**
10: **return** $\boldsymbol{x}_K$

---

## D  DISCUSSION ON CONDITIONING

**Projection type**  Methods that belong to this class aim to directly replace[1] what we have in the *range space* of the intermediate noisy $\boldsymbol{x}_i$ with the information from the measurement $\boldsymbol{y}$. Two representative works that utilize projection are Chung & Ye (2022) and Song et al. (2022). In Chung & Ye (2022), we use

$$\boldsymbol{x}_t = (\boldsymbol{I} - \boldsymbol{A}^\dagger \boldsymbol{A})\boldsymbol{x}_t' + \boldsymbol{A}^\dagger \boldsymbol{y}, \tag{72}$$

where the information from $\boldsymbol{y}$ will be used to fill in the range space of $\boldsymbol{A}^\dagger$. However, this is clearly problematic when considering the geometric viewpoint, as the sample may fall off the noisy manifold.

---

[1]Both hard (Chung & Ye, 2022) or soft (Song et al., 2022) constraints can be utilized.

---

**Algorithm 2** DDS (PI MRI; VP; noiseless)

---

**Require:** $\boldsymbol{\epsilon}_{\theta^*}, N, \{\alpha_t\}_{t=1}^N, \eta, \boldsymbol{A}, M$
1: $\boldsymbol{x}_N \sim \mathcal{N}(\boldsymbol{0}, \boldsymbol{I})$
2: **for** $t = N : 2$ **do**
3:      $\hat{\boldsymbol{\epsilon}}_t \leftarrow \boldsymbol{\epsilon}_{\theta^*}(\boldsymbol{x}_t)$
4:      $\triangleright$ Tweedie denoising
5:      $\hat{\boldsymbol{x}}_t \leftarrow (\boldsymbol{x}_t - \sqrt{1 - \bar{\alpha}_t}\hat{\boldsymbol{\epsilon}}_t)/\sqrt{\bar{\alpha}_t}$
6:      $\triangleright$ Data consistency
7:      $\hat{\boldsymbol{x}}_t' \leftarrow \texttt{CG}(\boldsymbol{A}^*\boldsymbol{A}, \boldsymbol{A}^*\boldsymbol{y}, \hat{\boldsymbol{x}}_t, M)$
8:      $\boldsymbol{\epsilon} \sim \mathcal{N}(\boldsymbol{0}, \boldsymbol{I})$
9:      $\triangleright$ DDIM sampling
10:      $\boldsymbol{x}_{t-1} \leftarrow \sqrt{\bar{\alpha}_{t-1}}\hat{\boldsymbol{x}}_t' - \sqrt{1 - \bar{\alpha}_{t-1} - \eta^2\tilde{\beta}_t^2}\hat{\boldsymbol{\epsilon}}_t + \eta\tilde{\beta}_t\boldsymbol{\epsilon}$
11: **end for**
12: $\boldsymbol{x}_0 \leftarrow (\boldsymbol{x}_1 - \sqrt{1 - \bar{\alpha}_1}\boldsymbol{\epsilon}_{\theta^*}(\boldsymbol{x}_1))/\sqrt{\bar{\alpha}_1}$
13: **return** $\boldsymbol{x}_0$

---

**Algorithm 3** DDS (PI MRI; VP; noisy)

---

**Require:** $\boldsymbol{\epsilon}_{\theta^*}, N, \{\alpha_t\}_{t=1}^N, \eta, \boldsymbol{A}, M, \gamma$
1: $\boldsymbol{x}_N \sim \mathcal{N}(\boldsymbol{0}, \boldsymbol{I})$
2: **for** $t = N : 2$ **do**
3:      $\hat{\boldsymbol{\epsilon}}_t \leftarrow \boldsymbol{\epsilon}_{\theta^*}(\boldsymbol{x}_t)$
4:      $\triangleright$ Tweedie denoising
5:      $\hat{\boldsymbol{x}}_t \leftarrow (\boldsymbol{x}_t - \sqrt{1 - \bar{\alpha}_t}\hat{\boldsymbol{\epsilon}}_t)/\sqrt{\bar{\alpha}_t}$
6:      $\triangleright$ Data consistency
7:      $\boldsymbol{A}_{\text{CG}} \leftarrow \boldsymbol{I} + \gamma\boldsymbol{A}^*\boldsymbol{A}$
8:      $\boldsymbol{y}_{\text{CG}} \leftarrow \hat{\boldsymbol{x}}_t + \gamma\boldsymbol{A}^*\boldsymbol{y}$
9:      $\hat{\boldsymbol{x}}_t' \leftarrow \texttt{CG}(\boldsymbol{A}_{\text{CG}}, \boldsymbol{y}_{\text{CG}}, \hat{\boldsymbol{x}}_t, M)$
10:      $\boldsymbol{\epsilon} \sim \mathcal{N}(\boldsymbol{0}, \boldsymbol{I})$
11:      $\triangleright$ DDIM sampling
12:      $\boldsymbol{x}_{t-1} \leftarrow \sqrt{\bar{\alpha}_{t-1}}\hat{\boldsymbol{x}}_t' - \sqrt{1 - \bar{\alpha}_{t-1} - \eta^2\tilde{\beta}_t^2}\hat{\boldsymbol{\epsilon}}_t + \eta\tilde{\beta}_t\boldsymbol{\epsilon}$
13: **end for**
14: $\boldsymbol{x}_0 \leftarrow (\boldsymbol{x}_1 - \sqrt{1 - \bar{\alpha}_1}\boldsymbol{\epsilon}_{\theta^*}(\boldsymbol{x}_1))/\sqrt{\bar{\alpha}_1}$
15: **return** $\boldsymbol{x}_0$

---

**Gradient type**  In the Bayesian reconstruction perspective, it is natural to incorporate the gradient of the likelihood as $\nabla \log p(\boldsymbol{x}_t|\boldsymbol{y}) = \nabla \log p(\boldsymbol{x}_t) + \nabla \log p(\boldsymbol{y}|\boldsymbol{x}_t)$. Here, while one can use $\nabla \log p(\boldsymbol{x}_t) \simeq \boldsymbol{s}_{\theta^*}(\boldsymbol{x}_t)$, $\nabla \log p(\boldsymbol{y}|\boldsymbol{x}_t)$ is intractable for all $t \neq 0$ (Chung et al., 2023a), one has to resort to approximations of $\nabla \log p(\boldsymbol{y}|\boldsymbol{x}_t)$. In a similar spirit to Score-MRI (Chung & Ye, 2022), one can simply use gradient steps of the form

$$\boldsymbol{x}_t = \boldsymbol{x}_t' - \xi_t \boldsymbol{A}^*(\boldsymbol{y} - \boldsymbol{A}\boldsymbol{x}_t'), \tag{73}$$

where $\xi_t$ is the step size (Jalal et al., 2021). Nevertheless, $\nabla_{\boldsymbol{x}_t} \log p(\boldsymbol{x}_t|\boldsymbol{y})$ is far from Gaussian as $i$ gets further away from 0, and hence is hard to interpret nor analyze what the gradient steps in the direction of $\boldsymbol{A}^*(\boldsymbol{y} - \boldsymbol{A}\boldsymbol{x}_t')$ is leading. Albeit not in the context of MRI, a more recent approach (Chung et al., 2023a) proposes to use

$$\boldsymbol{x}_t = \boldsymbol{x}_t' - \xi_i \nabla_{\boldsymbol{x}_{t+1}} \|\boldsymbol{y} - \boldsymbol{A}\hat{\boldsymbol{x}}_{t+1}\|_2^2. \tag{74}$$

As $\hat{\boldsymbol{x}}_{t+1}$ is the Tweedie denoised estimate and is free from Gaussian noise, (74) can be thought of as minimizing the residual *on the noiseless data manifold* $\mathcal{M}$. However, care must be taken since taking the gradient with respect to $\boldsymbol{x}_t$ corresponds to computing automatic differentiation through the neural net $\boldsymbol{s}_{\theta^*}$, often slowing down the compute by about $\times 2$ (Chung et al., 2023a).

**T2I vs. DIS**  For the former, the likelihood is usually given as a neural net-parameterized function $p_\phi(\boldsymbol{y}|\boldsymbol{x})$ (e.g. classifier, implicit gradient from CFG), whereas for the latter, the likelihood is given as an analytic distribution arising from some linear/non-linear forward operator (Kawar et al., 2022; Chung et al., 2023a).

---

**Algorithm 4** DDS (3D CT recon; VP)

---

**Require:** $\epsilon_{\theta^*}, N, \{\sigma_t\}_{t=1}^N, \eta, \boldsymbol{A}, M, \{\lambda_t\}_{t=1}^N$
1: $\boldsymbol{x}_N \sim \mathcal{N}(\boldsymbol{0}, \sigma_T^2 \boldsymbol{I})$
2: $\boldsymbol{z}_N \leftarrow \texttt{torch.zeros\_like}(\boldsymbol{x}_N)$
3: $\boldsymbol{w}_N \leftarrow \texttt{torch.zeros\_like}(\boldsymbol{x}_N)$
4: **for** $i = N : 2$ **do**
5: $\quad \hat{\boldsymbol{\epsilon}}_t \leftarrow \boldsymbol{\epsilon}_{\theta^*}(\boldsymbol{x}_t)$
6: $\quad \triangleright$ Tweedie denoising
7: $\quad \hat{\boldsymbol{x}}_t \leftarrow (\boldsymbol{x}_t - \sqrt{1 - \bar{\alpha}_t}\hat{\boldsymbol{\epsilon}}_t)/\sqrt{\bar{\alpha}_t}$
8: $\quad \triangleright$ Data consistency
9: $\quad \boldsymbol{A}_{\texttt{CG}} \leftarrow \boldsymbol{A}^T \boldsymbol{A} + \rho \boldsymbol{D}_z^T \boldsymbol{D}_z$
10: $\quad \boldsymbol{b}_{\texttt{CG}} \leftarrow \boldsymbol{A}^T \boldsymbol{y} + \rho \boldsymbol{D}_z^T (\boldsymbol{z}_{t+1} - \boldsymbol{w}_{t+1})$
11: $\quad \hat{\boldsymbol{x}}_t' \leftarrow \texttt{CG}(\boldsymbol{A}_{\texttt{CG}}, \boldsymbol{b}_{\texttt{CG}}, \hat{\boldsymbol{x}}_t, M)$
12: $\quad \boldsymbol{z}_t \leftarrow \mathcal{S}_{\lambda_t/\rho}(\boldsymbol{D}_z \hat{\boldsymbol{x}}_t' + \boldsymbol{w}_{t+1})$
13: $\quad \boldsymbol{w}_t \leftarrow \boldsymbol{w}_{t+1} + \boldsymbol{D}_z \hat{\boldsymbol{x}}_t' - \boldsymbol{z}_t$
14: $\quad \triangleright$ DDIM sampling
15: $\quad \boldsymbol{x}_{t-1} \leftarrow \sqrt{\bar{\alpha}_{t-1}}\hat{\boldsymbol{x}}_t' - \sqrt{1 - \bar{\alpha}_{t-1} - \eta^2 \tilde{\beta}_t^2}\hat{\boldsymbol{\epsilon}}_t + \eta \tilde{\beta}_t \boldsymbol{\epsilon}$
16: **end for**
17: $\boldsymbol{x}_0 \leftarrow (\boldsymbol{x}_1 - \sqrt{1 - \bar{\alpha}_1}\boldsymbol{\epsilon}_{\theta^*}(\boldsymbol{x}_1))/\sqrt{\bar{\alpha}_1}$

---

**Algorithm 5** DDS (PI MRI; VE)

---

**Require:** $\boldsymbol{s}_{\theta^*}, N, \{\sigma_t\}_{t=1}^N, \eta, \boldsymbol{A}, M$
1: $\boldsymbol{x}_N \sim \mathcal{N}(\boldsymbol{0}, \sigma_T^2 \boldsymbol{I})$
2: **for** $t = N : 2$ **do**
3: $\quad \hat{\boldsymbol{s}}_t \leftarrow \boldsymbol{s}_{\theta^*}(\boldsymbol{x}_t)$
4: $\quad \triangleright$ Tweedie denoising
5: $\quad \hat{\boldsymbol{x}}_t \leftarrow \boldsymbol{x}_t + \sigma_t^2 \boldsymbol{s}_{\theta^*}(\boldsymbol{x}_t)$
6: $\quad \triangleright$ Data consistency
7: $\quad \hat{\boldsymbol{x}}_t' \leftarrow \texttt{CG}(\boldsymbol{A}^* \boldsymbol{A}, \boldsymbol{A}^* \boldsymbol{y}, \hat{\boldsymbol{x}}_t, M)$
8: $\quad \boldsymbol{\epsilon} \sim \mathcal{N}(\boldsymbol{0}, \boldsymbol{I})$
9: $\quad \triangleright$ DDIM sampling
10: $\quad \boldsymbol{x}_{t-1} \leftarrow \hat{\boldsymbol{x}}_t' - \sigma_{t-1}\sigma_t \sqrt{1 - \tilde{\beta}^2 \eta^2}\hat{\boldsymbol{s}}_t + \sigma_{t-1}\boldsymbol{\epsilon}$
11: **end for**
12: $\boldsymbol{x}_0 \leftarrow \boldsymbol{x}_1 + \sigma_1^2 \boldsymbol{s}_{\theta^*}(\boldsymbol{x}_1)$
13: **return** $\boldsymbol{x}_0$

---

# E  ALGORITHMIC DETAILS

We provide the VP counterpart of the DDS VE algorithms presented in Algorithm 5,6 in Algorithm 2,4. The only differences are that the model is now parameterized with $\boldsymbol{\epsilon}_\theta$ that estimates the residual noise components and that we have a different noise schedule.

## E.1  ACCELERATED MRI

As stated in section 4.1, using $\geq 200$ NFE when using Algorithm 5 degrades the performance due to the numerical instability. To counteract this issue, we use the same iteration until $i \leq N/50 =: k$, and directly acquire the final reconstruction by Tweedie's formula:

$$\boldsymbol{x}_0 = \boldsymbol{x}_k + \sigma_k^2 \boldsymbol{s}_{\theta^*}(\boldsymbol{x}_k) \tag{75}$$

---

**Algorithm 6** DDS (3D CT recon; VE)

---

**Require:** $s_{\theta^*}, N, \{\sigma_t\}_{i=1}^N, \eta, \boldsymbol{A}, M, \{\lambda_t\}_{t=1}^N$
1: $\boldsymbol{x}_N \sim \mathcal{N}(\boldsymbol{0}, \sigma_T^2 \boldsymbol{I})$
2: $\boldsymbol{z}_N \leftarrow \texttt{torch.zeros\_like}(\boldsymbol{x}_N)$
3: $\boldsymbol{w}_N \leftarrow \texttt{torch.zeros\_like}(\boldsymbol{x}_N)$
4: **for** $t = N : 2$ **do**
5: $\quad \hat{\boldsymbol{s}}_t \leftarrow \boldsymbol{s}_{\theta^*}(\boldsymbol{x}_t)$
6: $\quad \triangleright$ Tweedie denoising
7: $\quad \hat{\boldsymbol{x}}_t \leftarrow \boldsymbol{x}_t + \sigma_t^2 \boldsymbol{s}_{\theta^*}(\boldsymbol{x}_t)$
8: $\quad \triangleright$ Data consistency
9: $\quad \boldsymbol{A}_{\mathsf{CG}} \leftarrow \boldsymbol{A}^T \boldsymbol{A} + \rho \boldsymbol{D}_z^T \boldsymbol{D}_z$
10: $\quad \boldsymbol{b}_{\mathsf{CG}} \leftarrow \boldsymbol{A}^T \boldsymbol{y} + \rho \boldsymbol{D}_z^T (\boldsymbol{z}_{t+1} - \boldsymbol{w}_{t+1})$
11: $\quad \hat{\boldsymbol{x}}'_t \leftarrow \mathsf{CG}(\boldsymbol{A}_{\mathsf{CG}}, \boldsymbol{b}_{\mathsf{CG}}, \hat{\boldsymbol{x}}_t, M)$
12: $\quad \boldsymbol{z}_t \leftarrow \mathcal{S}_{\lambda_t/\rho}(\boldsymbol{D}_z \hat{\boldsymbol{x}}'_t + \boldsymbol{w}_{t+1})$
13: $\quad \boldsymbol{w}_t \leftarrow \boldsymbol{w}_{t+1} + \boldsymbol{D}_z \hat{\boldsymbol{x}}'_t - \boldsymbol{z}_t$
14: $\quad \triangleright$ DDIM sampling
15: $\quad \boldsymbol{x}_{t-1} \leftarrow \hat{\boldsymbol{x}}'_t - \sigma_{t-1}\sigma_t\sqrt{1 - \tilde{\beta}^2\eta^2}\hat{\boldsymbol{s}}_t + \sigma_{t-1}\boldsymbol{\epsilon}$
16: **end for**
17: $\boldsymbol{x}_0 \leftarrow \boldsymbol{x}_1 + \sigma_1^2 \boldsymbol{s}_{\theta^*}(\boldsymbol{x}_1)$
18: **return** $\boldsymbol{x}_0$

---

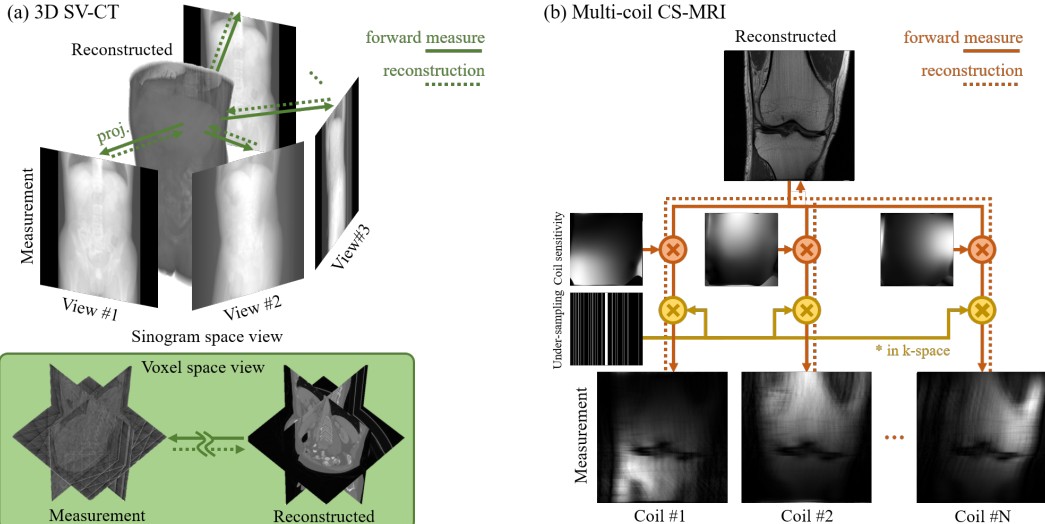

Figure 3: Illustration of the imaging forward model used in this work. (a) 3D sparse-view CT: the forward matrix $\boldsymbol{A}$ transforms the 3D voxel space into 2D projections. (b) Multi-coil CS-MRI: the forward matrix $\boldsymbol{A}$ first applies Fourier transform to turn the image into $k$-space. Subsequently, sensitivity maps are applied as element-wise product to achieve multi-coil measurements. Finally, the multi-coil measurements are sub-sampled with the masks.

## E.2 3D CT RECONSTRUCTION

Recall that to perform 3D reconstruction, we resort to the following optimization problem (omitting the indices for simplicity)

$$\hat{\boldsymbol{x}}^* = \arg\min_{\hat{\boldsymbol{x}}} \frac{1}{2}\|\boldsymbol{A}\hat{\boldsymbol{x}} - \boldsymbol{y}\|_2^2 + \lambda\|\boldsymbol{D}_z\hat{\boldsymbol{x}}\|_1. \tag{76}$$

One can utilize ADMM to stably solve the above problem. Here, we include the steps to arrive at Algorithms 6,4 for completeness. Reformulating into a constrained optimization problem, we have

$$\min_{\hat{\boldsymbol{x}},\boldsymbol{z}} \quad \frac{1}{2}\|\boldsymbol{y} - \boldsymbol{A}\hat{\boldsymbol{x}}\|_2^2 + \lambda\|\boldsymbol{z}\|_1 \tag{77}$$

$$\text{s.t.} \quad \boldsymbol{z} = \boldsymbol{D}_z\hat{\boldsymbol{x}}. \tag{78}$$

Then, ADMM can be implemented as separate update steps for the primal and the dual variables

$$\hat{\boldsymbol{x}}_{j+1} = \arg\min_{\hat{\boldsymbol{x}}_j} \frac{1}{2}\|\boldsymbol{y} - \boldsymbol{A}\hat{\boldsymbol{x}}_j\|_2^2 + \frac{\rho}{2}\|\boldsymbol{D}_z\hat{\boldsymbol{x}}_j - \boldsymbol{z}_j + \boldsymbol{w}\|_2^2 \tag{79}$$

$$\boldsymbol{z}_{j+1} = \arg\min_{\boldsymbol{z}_j} \lambda\|\boldsymbol{z}_j\|_1 + \frac{\rho}{2}\|\boldsymbol{D}_z\hat{\boldsymbol{x}}_j - \boldsymbol{z}_j + \boldsymbol{w}\|_2^2 \tag{80}$$

$$\boldsymbol{w}_{j+1} = \boldsymbol{w}_j + \boldsymbol{D}_z\hat{\boldsymbol{x}}_{j+1} - \boldsymbol{z}_{j+1}. \tag{81}$$

We have a closed-form solution for (79)

$$\hat{\boldsymbol{x}}_{j+1} = (\boldsymbol{A}^T\boldsymbol{A} + \rho\boldsymbol{D}_z^T\boldsymbol{D}_z)^{-1}(\boldsymbol{A}^T\boldsymbol{y} + \rho\boldsymbol{D}_z^T(\boldsymbol{z} - \boldsymbol{w})), \tag{82}$$

which can be numerically solved by iterative CG

$$\hat{\boldsymbol{x}}_{j+1} = \mathtt{CG}(\boldsymbol{A}_{\mathtt{CG}}, \boldsymbol{b}_{\mathtt{CG}}, \hat{\boldsymbol{x}}_j, M) \tag{83}$$

$$\boldsymbol{A}_{CG} := \boldsymbol{A}^T\boldsymbol{A} + \rho\boldsymbol{D}_z^T\boldsymbol{D}_z \tag{84}$$

$$\boldsymbol{b}_{CG} := \boldsymbol{A}^T\boldsymbol{y} + \rho\boldsymbol{D}_z^T(\boldsymbol{z} - \boldsymbol{w}). \tag{85}$$

Moreover, as (80) is in the form of proximal mapping (Parikh & Boyd, 2014), we have that

$$\boldsymbol{z}_{j+1} = \mathcal{S}_{\lambda/\rho}(\boldsymbol{D}_z\boldsymbol{x} + \boldsymbol{w}). \tag{86}$$

Thus, we have the following update steps

$$\hat{\boldsymbol{x}}_{j+1} = \mathtt{CG}(\boldsymbol{A}_{\mathtt{CG}}, \boldsymbol{b}_{\mathtt{CG}}, \hat{\boldsymbol{x}}_j, M)$$
$$\boldsymbol{z}_{j+1} = \mathcal{S}_{\lambda/\rho}(\boldsymbol{D}_z\hat{\boldsymbol{x}}_{j+1} + \boldsymbol{w}_j)$$
$$\boldsymbol{w}_{j+1} = \boldsymbol{w}_j + \boldsymbol{D}_z\hat{\boldsymbol{x}}_{j+1} - \boldsymbol{z}_{j+1}.$$

Usually, such an update step is performed in an iterative fashion until convergence. However, in our algorithm for 3D reconstruction (Algorithm 6, 4), we only use a single iteration of ADMM per each step of denoising. This is possible as we share the primal variable $\boldsymbol{z}$ and the dual variable $\boldsymbol{w}$ as global variables throughout the update steps, leading to proper convergence with a single iteration (fast variable sharing technique (Chung et al., 2023b)).

**Stabilizing Algorithm 6**   As stated in section 4.2, in order to stabilize Algorithm 6, we found it beneficial to first start the DC imposing strategy with simple CG update steps without TV prior, as used in our MRI experiments. We iterate our solver starting from $i = N$ down to $N/2$ using standard CG updates. Then, we switch to ADMM-TV scheme starting from $i = N/2 - 1$ down to 2.

# F   ADDITIONAL EXPERIMENTS

## F.1   NOISE OFFSET

For this experiment, we take 50 random proton density (PD) weighted images from the fastMRI validation dataset, and add Gaussian noise $\sigma_{\mathrm{GT}} = 7.00[\times 10^{-2}]$ to the images. For each noisy image, we apply the following DC step for each method

1. Score-MRI (Chung & Ye, 2022): (72)

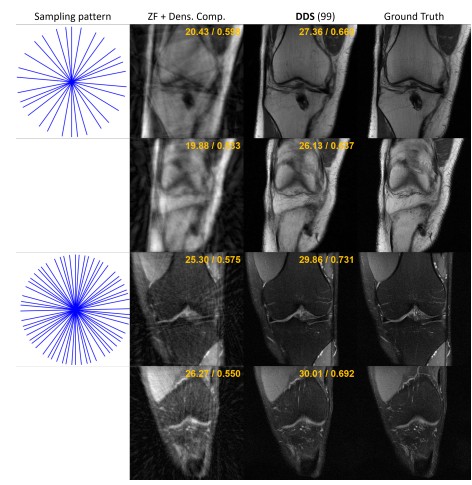

Figure 4: DDS reconstruction of CS-MRI on radial sampling trajectory. Col 1: sampling trajectory, 2: Zero filled reconstruction + density compensation, 3: DDS (99 NFE), 4: ground truth.

2. Jalal *et al.* (Jalal et al., 2021): (73) with step size as used in the implementation[2]

3. DPS (Chung et al., 2023a): (74) with step size 1.0

4. DDNM (Wang et al., 2023): (25)

5. DDS (CG): CG applied to the Tweedie denoised estimate, $M = 5$.

Once the update step is performed, the Gaussian noise level of the updated samples is estimated with the method from Chen et al. (2015). Note that as the estimation method is imperfect, there is already a gap between the ground truth noise level and the estimated noise level.

| Method | No process | Score-MRI | Jalal *et al.* | DPS | DDNM | **DDS (CG)** |
|---|---|---|---|---|---|---|
| $\sigma_{\text{est}}$ | 7.556 | 5.959 | 6.303 | 8.527 | 8.256 | 7.859 |
| $\|\sigma_{\text{est}}^{\text{np}} - \sigma_{\text{est}}\|$ | 0.000 | 1.597 | 1.253 | 0.917 | 0.700 | **0.303** |

Table 5: Noise offset experiment. Gaussian noise level estimated with Chen et al. (2015). Real noise level: $\sigma_{\text{GT}} = 7.00[\times 10^{-2}]$; $\sigma_{\text{est}}^{\text{np}} = 7.56[\times 10^{-2}]$

## F.2 NON-CARTESIAN MRI TRAJECTORIES

To put an emphasis on the fact that the proposed method is agnostic to the forward imaging model at hand, for the first time in DIS literature, we conduct MRI reconstruction experiments on the non-Cartesian trajectory, which involves non-uniform Fast Fourier Transform (NUFFT) (Fessler & Sutton, 2003). In Fig. 4 we show that DDS is able to reconstruct high-fidelity images from radial trajectory sampling, even under aggressive accelerations.

## F.3 STOCHASTICITY IN SAMPLING

For our sampling strategy, the stochasticity of the sampling process is determined by the parameter $\eta \in [0, 1]$. When $\eta \to 0$, the sampling becomes deterministic, whereas when $\eta \to 1$, the sampling becomes maximally stochastic. It is known in the literature that for unconditional sampling using low NFE, setting $\eta$ as close to 0 leads to better performance Song et al. (2021a). In Fig. 6, we see a similar trend when we set NFE = 20. However, when we set NFE $\geq 50$, we observe that the choice of $\eta$ does not matter too much, as it often fluctuates within the boundary that can be as well be thought of as arising from the inherent stochasticity of the sampling procedure. This is different from the observation made in (Song et al., 2021a), which can be thought of as stemming from the conditional sampling strategy that the proposed method uses.

## F.4 INSTABILITY IN VE PARAMETERIZATION

| NFE | 4000 | 500 | 200 | 100 | 50 | 30 |
|---|---|---|---|---|---|---|
| Score-MRI Chung & Ye (2022) | **33.25** | 33.19 | 33.13 | 31.67 | 3.015 | 3.239 |
| Ours | 32.07 | 31.16 | 31.99 | **33.69** | 31.79 | 30.40 |

Table 6: PSNR [db] of uniform 1D $\times 4$ acc. reconstruction with varying NFEs.

One observation that is made in this experiment, however, is that using $\geq 200$ NFEs for the proposed method *degrades* the performance. We find that this degradation stems from the numerical pathologies that arise when VE-SDE is combined with the parameterizing the neural network with $s_\theta$. Specifically, the score function is parameterized to estimate $s_{\theta^*}(\boldsymbol{x}_t) \simeq -\frac{\boldsymbol{x}_t - \boldsymbol{x}_0}{\sigma_t^2} = \boldsymbol{\epsilon}/\sigma_t$. Near $t = 0$, $\sigma_t$ attains a

---

[2]https://github.com/utcsilab/csgm-mri-langevin/blob/main/main.py

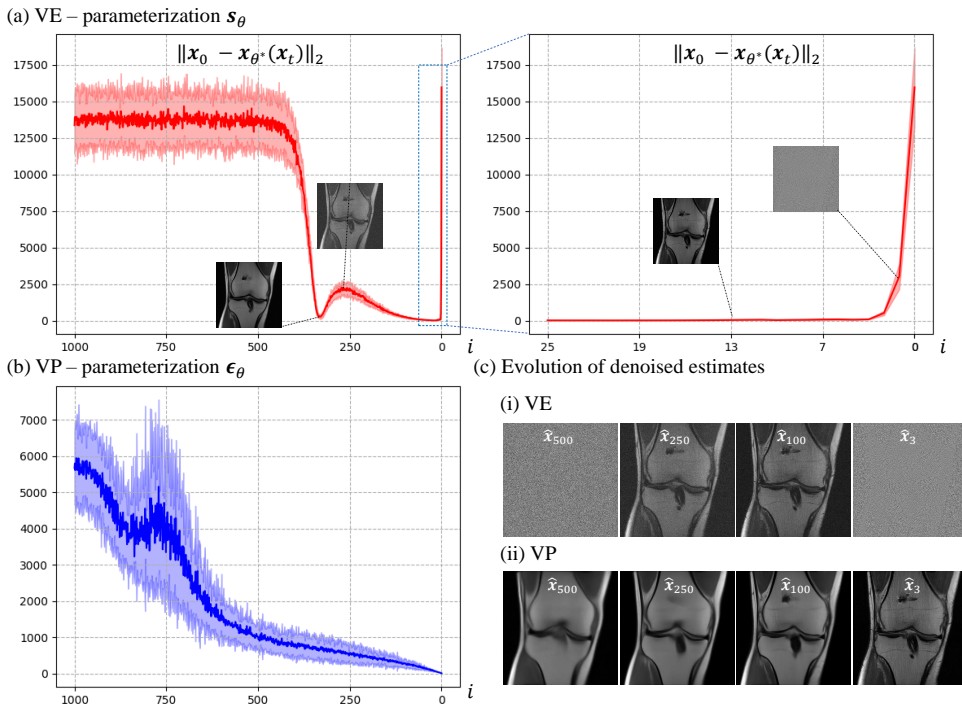

(a) VE – parameterization $s_\theta$

(b) VP – parameterization $\epsilon_\theta$

(c) Evolution of denoised estimates

(i) VE

(ii) VP

Figure 5: Evolution of the reconstruction error through time. $\pm 1.0\sigma$ plot. (a) VE parameterized with $s_\theta$, (b) VP parameterized with $\epsilon_\theta$, (c) Visualization of $\hat{x}_t$.

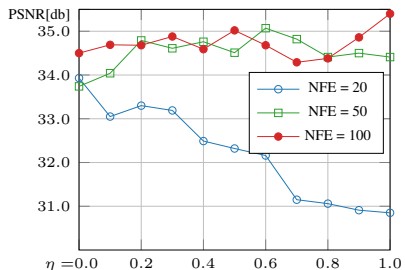

Figure 6: Ablation study on the selection of $\eta$ for Algorithm 2.

Table 7: Quantitative evaluation of LA-CT (90°) on the AAPM 256×256 test set. **Bold**: Best.

| Method | Axial* | | Coronal | | Sagittal | |
|---|---|---|---|---|---|---|
| | PSNR ↑ | SSIM ↑ | PSNR ↑ | SSIM ↑ | PSNR ↑ | SSIM ↑ |
| DDS VP (49) | **35.07** | **0.963** | **32.90** | **0.955** | **29.34** | **0.861** |
| DiffusionMBIR Chung et al. (2023b) | 34.92 | 0.956 | 32.48 | 0.947 | 28.82 | 0.832 |
| MCG Chung et al. (2022a) | 26.01 | 0.838 | 24.55 | 0.823 | 21.59 | 0.706 |
| Song et al.Song et al. (2022) | 27.80 | 0.852 | 25.69 | 0.869 | 22.03 | 0.735 |
| DPS Chung et al. (2023a) | 25.32 | 0.829 | 24.80 | 0.818 | 21.50 | 0.698 |
| Lahiri et al. Lahiri et al. (2022) | 28.08 | 0.931 | 26.02 | 0.856 | 23.24 | 0.812 |
| Zhang et al. Zhang et al. (2017) | 26.76 | 0.879 | 25.77 | 0.874 | 22.92 | 0.841 |
| ADMM-TV | 23.19 | 0.793 | 22.96 | 0.758 | 19.95 | 0.782 |

| | 8-view | | | | | | 4-view | | | | | | 2-view | | | | | |
|---|---|---|---|---|---|---|---|---|---|---|---|---|---|---|---|---|---|---|
| | Axial* | | Coronal | | Sagittal | | Axial* | | Coronal | | Sagittal | | Axial* | | Coronal | | Sagittal | |
| Method | PSNR ↑ | SSIM ↑ | PSNR ↑ | SSIM ↑ | PSNR ↑ | SSIM ↑ | PSNR ↑ | SSIM ↑ | PSNR ↑ | SSIM ↑ | PSNR ↑ | SSIM ↑ | PSNR ↑ | SSIM ↑ | PSNR ↑ | SSIM ↑ | PSNR ↑ | SSIM ↑ |
| DDS VP (19) | 1.38 | 0.05 | 1.51 | 0.06 | 1.76 | 0.06 | 1.96 | 0.07 | 1.26 | 0.10 | 2.73 | 0.11 | 2.01 | 0.08 | 2.89 | 0.10 | 2.01 | 0.21 |
| DDS VP (49) | 1.96 | 0.08 | 1.93 | 0.10 | 1.34 | 0.05 | 2.01 | 0.12 | 2.10 | 0.11 | 1.96 | 0.10 | 2.90 | 0.23 | 2.08 | 0.17 | 2.78 | 0.19 |
| DDS VE (99) | 1.75 | 0.09 | 1.47 | 0.11 | 1.55 | 0.10 | 1.98 | 0.13 | 2.22 | 0.07 | 1.80 | 0.15 | 2.52 | 0.19 | 2.90 | 0.17 | 2.75 | 0.20 |
| DiffusionMBIR (Chung et al., 2023b) (4000) | 1.50 | 0.09 | 1.37 | 0.07 | 1.93 | 0.08 | 1.83 | 0.13 | 1.70 | 0.15 | 1.28 | 0.10 | 1.58 | 0.13 | 2.37 | 0.15 | 2.58 | 0.20 |
| DPS (Chung et al., 2023a) (1000) | 1.95 | 0.19 | 2.05 | 0.14 | 2.21 | 0.17 | 1.71 | 0.16 | 2.33 | 0.17 | 2.00 | 0.12 | 3.01 | 0.21 | 3.29 | 0.19 | 3.57 | 0.29 |
| Score-Med (Song et al., 2022) (4000) | 1.68 | 0.15 | 1.57 | 0.18 | 1.88 | 0.24 | 2.14 | 0.13 | 2.40 | 0.15 | 3.07 | 0.22 | 2.69 | 0.29 | 2.50 | 0.19 | 3.22 | 0.19 |
| MCG (Chung et al., 2022a) (4000) | 1.55 | 0.15 | 1.60 | 0.14 | 1.61 | 0.14 | 1.93 | 0.20 | 1.70 | 0.18 | 1.88 | 0.17 | 2.13 | 0.16 | 2.66 | 0.19 | 2.78 | 0.27 |
| Lahiri et al. (2022) | 1.28 | 0.12 | 1.52 | 0.10 | 2.08 | 0.13 | 1.30 | 0.17 | 1.57 | 0.10 | 1.44 | 0.12 | 2.40 | 0.15 | 2.31 | 0.22 | 3.07 | 0.24 |
| FBPConvNet (Jin et al., 2017) | 2.05 | 0.24 | 3.74 | 0.15 | 3.45 | 0.29 | 3.67 | 0.31 | 3.51 | 0.30 | 3.33 | 0.27 | 3.17 | 0.25 | 3.04 | 0.31 | 4.07 | 0.22 |
| ADMM-TV | 2.73 | 0.16 | 2.57 | 0.18 | 2.81 | 0.16 | 2.90 | 0.25 | 2.88 | 0.16 | 3.44 | 0.25 | 3.01 | 0.22 | 2.92 | 0.19 | 2.70 | 0.16 |

Table 8: Standard deviation of the quantitative metrics presented in Table. 4, which correspond to the results for sparse-view CT reconstruction. Mean values in Tab. 4. (Numbers in parenthesis): NFE.

very small value $e.g.\sigma_0 = 0.01$ (Kim et al., 2022), meaning that the score function has to approximate relatively large values in such regime, leading to numerical instability. This phenomenon is further illustrated in Fig. 5 (a), where the reconstruction (i.e. denoising) error has a rather odd trend of jumping up and down, and completely diverging as $t \to 0$. This may be less of an issue for using samplers such as PC where $\hat{x}_i$ are not directly used but becomes a much bigger problem when

the proposed sampler is used. In fact, for NFE > 200, we find that simply truncating the last few evolutions is necessary to yield the result reported in Tab. 6 (See appendix E.1 for details).

Such instabilities worsened when we tried scaling our experiments to complex-valued PI reconstruction due to the network only being trained on magnitude images. On the other hand, the reconstruction errors for VP trained with epsilon matching have a much stabler evolution of denoising reconstructions, suggesting that it is indeed a better fit in the context of the proposed methodology. Hence, all experiments reported hereafter use a network parameterized with $\epsilon_\theta$ trained within a VP framework, and also by stacking real/imag part in the channel dimension to account for the complex-valued nature of the MR imagery and to avoid using $\times 2$ NFE for a single level of denoising.

# G EXPERIMENTAL DETAILS

## G.1 DATASETS

**fastMRI knee.** We conduct all PI experiments with fastMRI knee dataset (Zbontar et al., 2018). Following the practices from Chung & Ye (2022), among the 973 volumes of training data, we drop the first/last five slices from each volume. As the test set, we select 10 volumes from the validation dataset, which consists of 281 2D slices. While Chung et al. (2022b) used DICOM magnitude images to train the network, we used the minimum-variance unbiased estimates (MVUE) (Jalal et al., 2021) complex-valued data by stacking the real and imaginary parts of the images into the channel dimension. When performing inference, we pre-compute the coil sensitivity maps with ESPiRiT (Uecker et al., 2014).

**AAPM.** AAPM 2016 CT low-dose grand challenge data leveraged in Chung et al. (2022a; 2023b) is used. From the filtered backprojection (FBP) reconstruction of size $512 \times 512$, we resize all the images to have the size $256 \times 256$ in the axial dimension. We use the same 1 volume of testing data that was used in Chung et al. (2023b). This corresponds to 448 axial slices, 256 coronal, and 256 sagittal slices in total. To generate sparse-view and limited angle measurements, we use the parallel view geometry for simplicity, with the `torch-radon` (Ronchetti, 2020) package.

## G.2 NETWORK TRAINING

VE models for both MRI/CT are taken from the official github repositories (Chung & Ye, 2022; Chung et al., 2022a), which are both based on `ncsnpp` model of Score-SDE (Song et al., 2021b). In order to train our VP epsilon matching models, we take the U-Net implementation from ADM (Dhariwal & Nichol, 2021), and train each model for 1M iterations with the batch size of 4, initial learning rate of $1e - 4$ on a single RTX 3090 GPU. Training took about 3 days for each task.

## G.3 REJECTION SAMPLING

When running Algorithm 2 in the low NFE regime (e.g. 19, 49), we see that our method sometimes yields infeasible reconstructions. Sampling 100 reconstructions for 1D uniform $\times 4$ acc., we see that 5% of the samples were infeasible for 19 NFE, and 3% of the samples were infeasible for 49 NFE. In such cases, we simply compute the Euclidean norm of the residual $\|\boldsymbol{y} - \boldsymbol{A}\hat{\boldsymbol{x}}\|$ with the reconstructed sample $\hat{\boldsymbol{x}}$, and reject the sample if the residual exceeds some threshold value $\tau$. Even when we consider the additional time costs that arise from re-sampling the rejected reconstructions, we still achieve dramatic acceleration to previous methods Song et al. (2022); Chung & Ye (2022).

## G.4 COMPARISON METHODS

### G.4.1 ACCELERATED MRI

**Score-MRI (Chung & Ye, 2022)** We use the official pre-trained model[3] with 2000 PC sampling. Note that for PI, this amounts to running the sampling procedure per coil. When reducing the number of NFE presented in Fig. 1, we use linear discretization with wider bins.

---

[3]https://github.com/HJ-harry/score-MRI

**Jalal et al. (2021).** As the official pre-trained model is trained on fastMRI brain MVUE images, in order to perform fair comparison, we train the NCSN v2 model with the same fastMRI knee MVUE images for 1M iterations in the setting identical to when training the model for the proposed method. For inference, we follow the default annealing step sizes as proposed in the original paper. We use 3 Langevin dynamics steps per noise scale for 700 discretizations, which amounts to a total of 2100 NFE. When reducing the number of NFE presented in Fig. 1, we keep the 3 Langevin steps intact, and use linear discretization with wider bins.

**E2E-Varnet (Sriram et al., 2020), U-Net.** We train the supervised learning-based methods with Gaussian 1D subsampling as performed in Chung & Ye (2022), adhering to the official implementation and the default settings of the original work.

**TV.** We use the implementation in `sigpy.mri.app.TotalVariation`[4], with calibrated sensitivity maps with ESPiRiT (Uecker et al., 2014). The parameters for the optimizer are found via grid search on 50 validation images.

### G.4.2   3D CT RECONSTRUCTION

**DiffusionMBIR (Chung et al., 2023b), MCG (Chung et al., 2022a).** Both methods use the same score function as provided in the official repository[5]. For both DiffusionMBIR and Score-CT, we use 2000 PC sampler (4000 NFE). For DiffusionMBIR, we set $\lambda = 10.0, \rho = 0.04$, which is the advised parameter setting for the AAPM 256×256 dataset. For Chung *et al.*, we use the iterating ART projections as used in the comparison study in Chung et al. (2023b).

**Lahiri et al. (2022).** We use two stages of 3D U-Net based CNN architectures. For each greedy optimization process, we train the network for 300 epochs. For CG, we use 30 iterations at each stage. Networks were trained with the Adam optimizer with a static learning rate of $1e - 4$, batch size of 2.

**FBPConvNet Jin et al. (2017), Zhang et al. (2017).** Both methods utilize the same network architecture and the same training strategy, only differing in the application (SV, LA). We use a standard 2D U-Net architecture and train the models with 300 epochs. Networks were trained with the Adam optimizer with a learning rate of $1e - 4$, batch size 8.

**ADMM-TV.** Following the protocol of Chung et al. (2023b), we optimize the following objective

$$x^* = \arg\min_{x} \frac{1}{2}\|Ax - y\|_2^2 + \lambda\|Dx\|_{2,1}, \tag{87}$$

with $D := [D_x, D_y, D_z]$, and is solved with standard ADMM and CG. Hyper-parameters are set identical to Chung et al. (2023b).

## H   QUALITATIVE RESULTS

---

[4]https://github.com/mikgroup/sigpy
[5]https://github.com/HJ-harry/MCG_diffusion

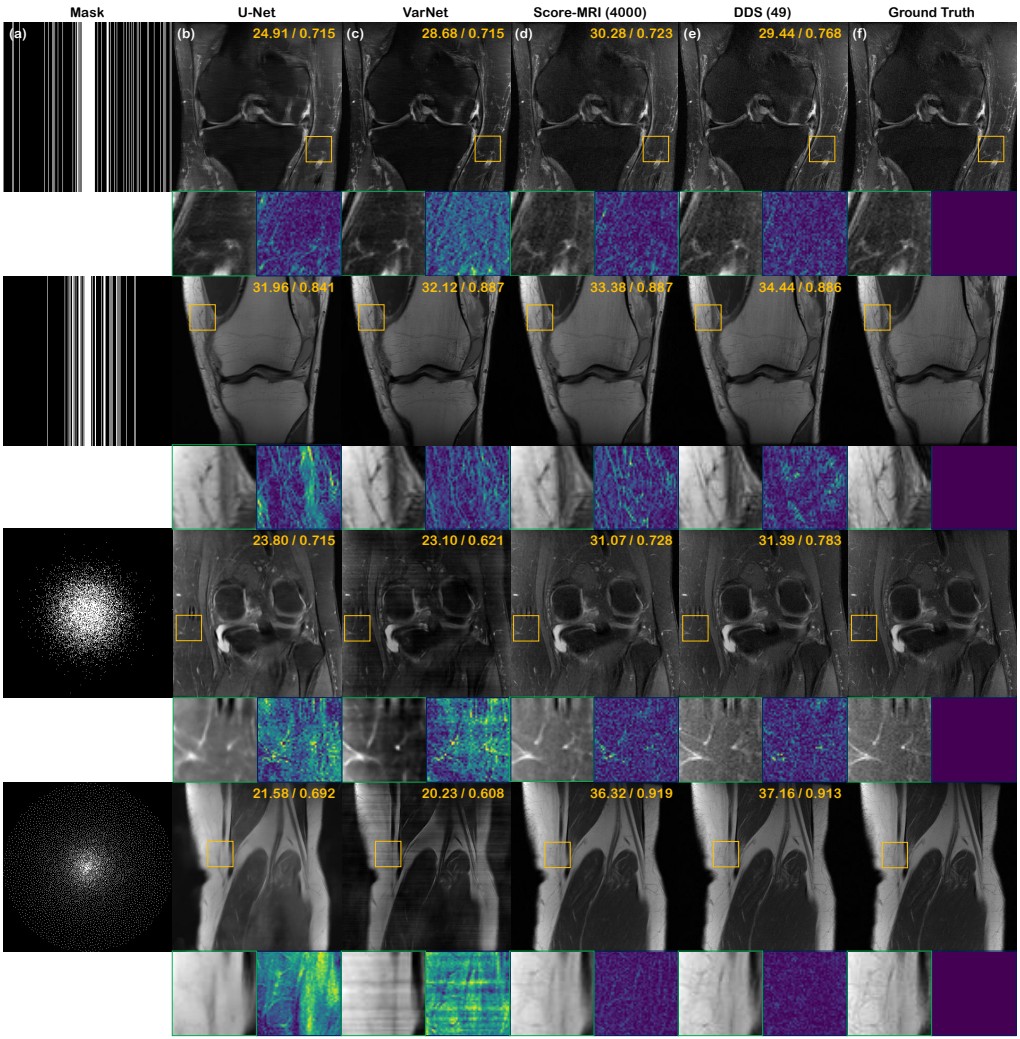

Figure 7: Comparison of parallel imaging reconstruction results. (a) subsampling mask (1st row: uniform1D×4, 2nd row: Gaussian1D×8, 3rd row: Gaussian2D×8, 4th row: variable density poisson disc ×8), (b) U-Net (Zbontar et al., 2018), (c) E2E-VarNet (Sriram et al., 2020), (d) Score-MRI (Chung & Ye, 2022) ($4000 \times 2 \times c$ NFE), (e) DDS (49 NFE), (f) ground truth.

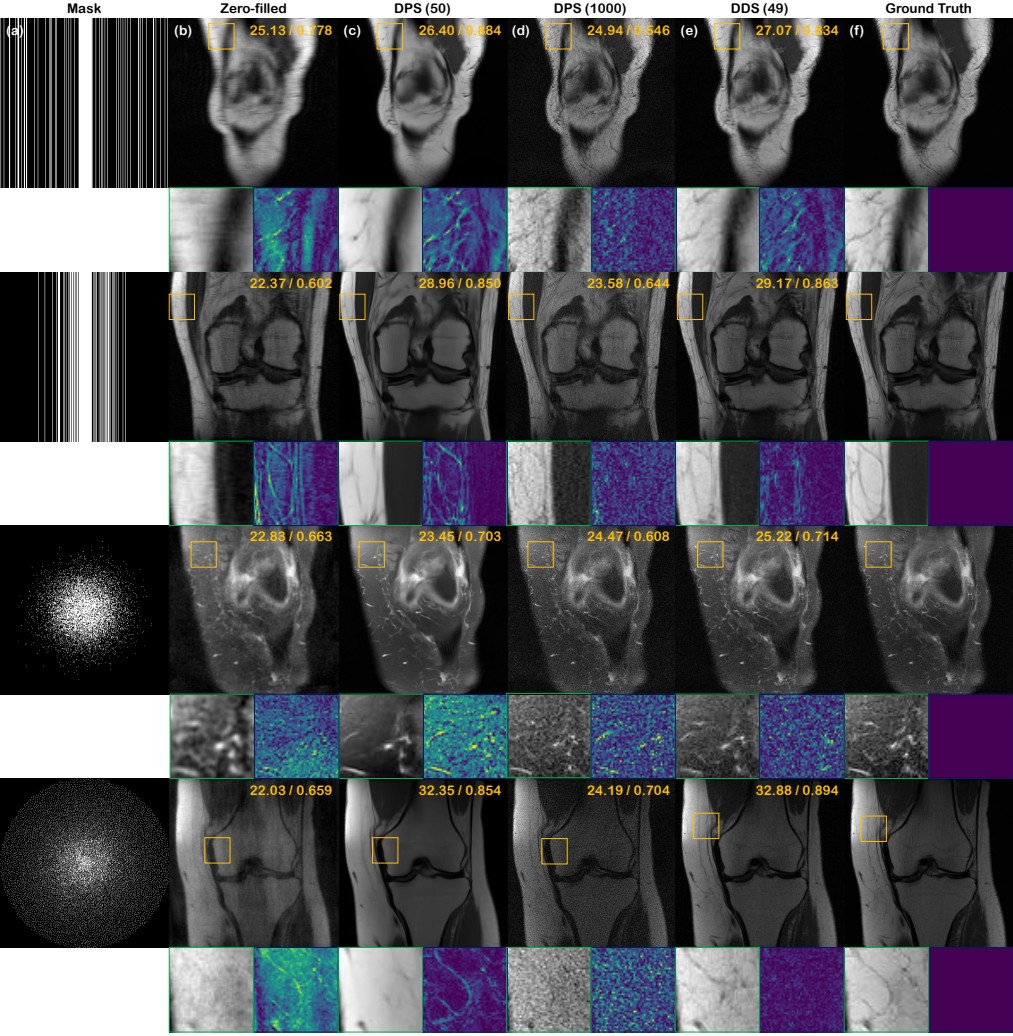

Figure 8: Comparison of noisy ($\sigma = 0.05$) parallel imaging reconstruction results. (a) subsampling mask (1st row: uniform1D×4, 2nd row: Gaussian1D×4, 3rd row: Gaussian2D×8, 4th row: variable density poisson disc ×8), (b) Zero-filled, (c) DPS (Chung et al., 2023a) (50 NFE), (d) DPS (Chung et al., 2023a) (1000 NFE), (e) DDS (49 NFE), (f) ground truth. Numbers in the top right corners denote PSNR and SSIM, respectively.

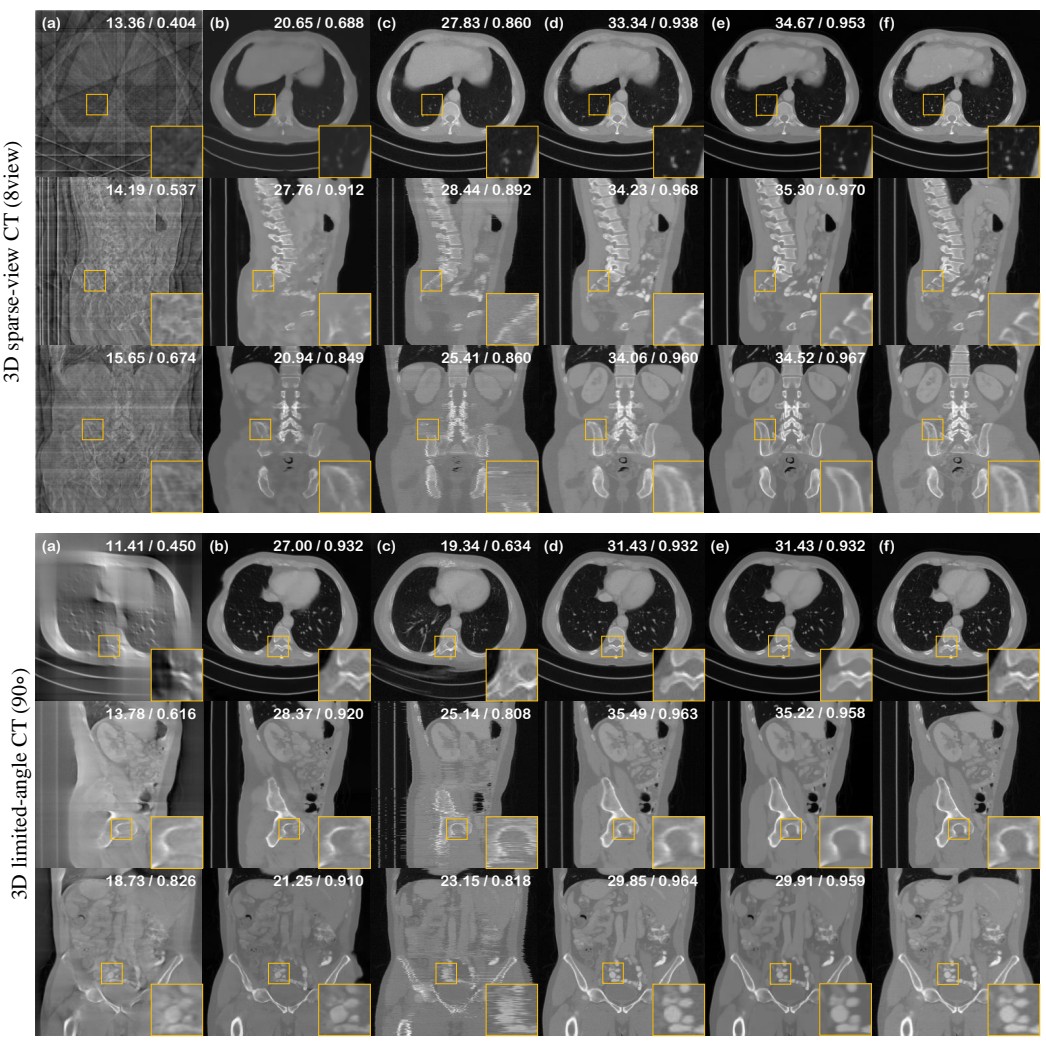

Figure 9: Comparison of 3D CT reconstruction results. (**Top**): 8-view SV-CT, (**Bottom**): 90° LA-CT. (a) FBP, (b) Lahiri *et al.*, (c) Chung *et al.*, (d) DiffusionMBIR (4000 NFE), (e) DDS (49 NFE), (f) ground truth. Numbers in top right corners denote PSNR and SSIM, respectively.

