# OpenReview forum: "Decomposed Diffusion Sampler for Accelerating Large-Scale Inverse Problems"
_ICLR.cc/2024/Conference — ICLR 2024 poster_

### Official Review · Reviewer_YQ7m · 2023-10-19

**Soundness:** 3 good
**Presentation:** 3 good
**Contribution:** 3 good
**Rating:** 8
**Confidence:** 4

**Summary:**

In this paper, the authors propose to integrate classical Krylov subspace methods, widely used in large linear inverse problems, with state-of-the-art (SOTA) diffusion models applied to medical image reconstruction. An efficient diffusion sampling strategy is proposed that, if the tangent space at a denoised sample forms a Krylov subspace, allows the Conjugate Gradient (CG) method to maintain data consistency updates within this subspace. This eliminates the need for the computationally expensive manifold-constrained gradient (MCG) reducing computation time. The approach achieves  SOTA reconstruction quality in large scale inverse imaging problems, such as multi-coil MRI and 3D CT reconstruction.

**Strengths:**

•	The idea of applying Krylov subspace methods for efficient sampling strategies in stable diffusion models is novel and significant since practical applications such as medical image reconstructions are usually large inverse problems.

•	The organization of the paper is clear with useful introductions to the relevant topics such as diffusion models and Krylov subspace methods.

•	Comparison with various SOTA methods is provided showing that the proposed method achieves higher quality results with significant fewer number of iterations.

**Weaknesses:**

Main weaknesses:

•	The comparison provided primarily centers on state-of-the-art (SOTA) diffusion models for reconstructing images from undersampled data and only a comparison with a the naïve Total Variation (TV) is included. However, I believe it lacks a comparison with the classical Compressed Sensing (CS) approach, which relies on sparse image representations and has shown success, especially in MRI applications (as in the classical reference [*]). It's unclear whether SOTA diffusion models can compete with traditional CS algorithms. Perhaps referencing recent studies that compare diffusion models with classical CS methods would offer a more comprehensive perspective on the field's current status.

•	It is not clear why the Krylov method, originally developed for square matrix A can be successfully applied to the image reconstruction problem when A has fewer rows than columns. See my question number 1 below.

•	The proposed method has a limitation since it only considers linear forward models with the form y=Ax (see my question number 2 below).

•	The methods requires that the tangent space on the clean manifold at a denoised sample forms a Krylov subspace and experimental results suggests that such assumption is met. However, an intuition about why the tangent space forms a Krylov subspace is missing. It would be useful to have identified conditions for such assumption become true (see my question number 3 below).

•	Experiments: Quality results are expressed in terms of PSNR and SSIM values, but it is not clear if these are averaged values and the number of instances considered in the corresponding average. The experiments should also report variability of the results, for example showing standard deviation for a more comprehensive comparison of different methods (See my question number 4).

Minor comments:

-	Fig. 2 caption should indicate what are the numbers at the bottom of each subimage, which I think they are PSNR/SSIM

-	Acronym NFE is not defined in the paper, please introduce its definition (Number of Forward Evaluations?)

-	Just before eq. (3): “two subspace K and L” -> “two subspaces K and L”

-	Just after eq. (7): “for given the optimization problem” -> “for a given the optimization problem”

-	Page 6: “guranteed” -> “guaranteed”

-	Problem setting considers only the noiseless case (equation (29)), however experiments in Fig. 2 shows some results in the noisy case. Please include the noisy case in the problem setting.

-	In the problem setting, s^{(1)}, s^{(2)}, …, s^{(c)} are not defined. For single-coil case s^{(1)}=I but it is not clear how they are defined in the multi-coil case. Please, clarify.

-	PC sampler is not defined.

[*] Lustig, M., Donoho, D. and Pauly, J.M. (2007), Sparse MRI: The application of compressed sensing for rapid MR imaging. Magn. Reson. Med., 58: 1182-1195. https://doi.org/10.1002/mrm.21391

**Questions:**

1.	In section 2, it is stated that in the forward model (1): y=Ax, we can assume A square without losing generality because, if not, we can consider the modified equation (2): A*y=A*Ax, because its solution is also a solution of problem (1). However, equation (1) has infinite number of solutions when A has fewer rows than columns as in the considered in the image reconstruction scenarios, and the solution of eq. (2) is unique if A is full rank matrix. It is not clear why the unique solution of eq. (2) is the most preferable solution over all possible solutions of eq. (1). Could you please clarify about this?

2.	The proposed method has a limitation since it only considers linear forward models with the form y=Ax and uses Krylov method to minimize l(x) = ||y - Ax||^2. However, just after eq. (6) it is mentioned that Krylov method can be extended to non-linear problems and the example l(x) = 0.5||y - Ax||^2 is given, which is confusing since it is the same optimization problem as in the linear model y=Ax. Can you clarify what do you mean by extending Krylov method to nonlinear problems? Can thus you apply your method to an inverse problem having a nonlinear forward model?

3.	The methods requires that the tangent space on the clean manifold at a denoised sample forms a Krylov subspace and experimental results suggests that such assumption is met. What is the intuition behind this assumption? Can you provide some conditions for such assumption become true?

4.	Regarding the reported PSNR and SSIM values: Are these average values? If so, how many samples have you consider computing the average? Can you also report the variability of these measures to derive conclusions about the comparison of methods? Are the differences between methods statistically significative in some sense?

---

> ### Author Response · Authors · 2023-11-19
> **Reply to reviewer YQ7m**
>
> **W1. Are DIS better than classic CS methods? References?**
>
> **A.** Thank you for the suggestion. Please see modified section 4.1., where we cite two works on using diffusion model-based inverse problem solvers, which show that the proposed methods largely outperform the classic CS algorithms such as L1-wavelet [1] and L1-ESPiRiT [2]. While we do not directly compare ours against these CS baselines, we do compare against [3], implying superiority.
>
> **W2, Q1. Relation between the solution to $\mathbf{y} = \mathbf{A}\mathbf{x}$ and the modified equation by left multiplying $\mathbf{A}^\top$**
>
> **A.** Thanks for your careful observation. We agree that even after using $\mathbf{A}^*\mathbf{y} = \mathbf{A}^*\mathbf{A}\mathbf{x}$ to obtain a square matrix for Krylov method, the solution is not unique. This is why the notion of Krylov subspace is important. By enforcing that the solution lies in a specific lower-dimensional Krylov subspace whose dimension is less than equal to $\text{rank}(\mathbf{A}^*\mathbf{A})$, the ill-posedness of the zero-finding in $\mathbf{A}^*\mathbf{y} = \mathbf{A}^*\mathbf{A}\mathbf{x}$ can be overcome.
>
> **W3, Q2. The proposed method can be applied to linear models. Can you clarify what do you mean by extending Krylov method to nonlinear problems?**
>
> **A.** Apologies for the confusion. When solving problems with a nonlinear operator $\mathcal{A}(\cdot)$ we can change the objective to $\|\|\mathbf{y} - \mathcal{A}(\mathbf{x})\|\|_2^2/2$. Then, one can use methods such as Newton-Krylov method to linearize the given problem with Taylor approximation, and apply standard Krylov method in that approximate regime. We have modified our manuscript to elaborate without experiments as this is beyond the scope of this work.
>
> **W4, Q3. What is the intuition behind the assumption that the tangent of the data manifold forms a Krylov subspace?**
>
> **A.** In practical use of Krylov subspace in linear inverse solver, the approach becomes powerful and require less CG optimization step especially when the initialization is good. This is because the solution may lie in lower dimensional affine space around the initialization point.
>
> Our diffusion inverse solvers combined with Tweedie’s denoising step gives quite a good approximation of the true solution especially when the reverse sampling progresses, so that CG may achieve a good solution with a small number of iteration. Our intuition is to find an iterative solver to satisfy the data consistency with the smallest number of iterations in order to avoid the results from falling off the correction diffusion manifold. Empirically we found that CG was one of the most powerful solvers with the initialization with diffusion solver, which led us to believe that the local tangent space may be in the form of the Krylov subspace.
>
> We believe that the empirical study in Section F.1. does indeed show that our assumption is close to reality. That said, it would be definitely interesting to further investigate on when the assumptions will perfectly hold as a future direction of research.
>
> **W5, Q4. Experiments: how many slices were considered? Variability of the measures and their statistical significance?**
>
> **A.** For CS-MRI, the total number of test slices is 292. For CT, it is 448, 256, 256 slices for axial, coronal, and sagittal slices, respectively. The results reported in the Tables are the average values. For completeness, we also included the std values in Tab. 1. Due to limited space, we included the std values for Tab. 4 in Appendix Tab. 8. We do not perform statistical tests as it would require pairwise comparison of DDS against all the baseline methods.
>
> **MC. Fig 2 caption should indicate what the numbers mean in the image**
>
> **A.** Thank you for the careful reading. Fixed.
>
>
> **References**
>
> [1] Lustig, M., Donoho, D. and Pauly, J.M. (2007), Sparse MRI: The application of compressed sensing for rapid MR imaging. Magn. Reson. Med., 58: 1182-1195. https://doi.org/10.1002/mrm.21391
>
> [2] Uecker, Martin, et al. "ESPIRiT—an eigenvalue approach to autocalibrating parallel MRI: where SENSE meets GRAPPA." Magnetic resonance in medicine 71.3 (2014): 990-1001.
>
> [3] Jalal, Ajil, et al. "Robust compressed sensing MRI with deep generative priors." NeurIPS 2021.

---

> > ### Comment · Reviewer_YQ7m · 2023-11-22
> > **score updated**
> >
> > Dear authors,
> >
> > Thanks for addressing my questions. I am satisfied with your answers and believe that the paper has improved so I decided to increase my score.
> >
> > Best

---

> > > ### Author Response · Authors · 2023-11-23
> > > **Thanks!**
> > >
> > > Dear Reviewer YQ7m,
> > >
> > > Thanks for the positive comments and raising the score.  We are happy to hear that you are satisfied with our answers and that our paper has improved.

---

### Official Review · Reviewer_tJRQ · 2023-10-30

**Soundness:** 3 good
**Presentation:** 3 good
**Contribution:** 3 good
**Rating:** 6
**Confidence:** 4

**Summary:**

This paper proposes a novel and efficient diffusion sampling strategy that synergistically combine the diffusion sampling and Krylov subspace methods.
Specifically, it prove that if the tangent space at a denoised sample by Tweedie’s formula forms a Krylov subspace, then the CG initialized with the denoised data ensures the data consistency update to remain in the tangent space.
This eliminates the necessity of computing the manifold-constrained gradient (MCG), leading to a more efficient diffusion sampling method.
The proposed method achieves state-of-the-art  reconstruction quality on challenging real-world medical inverse imaging problems, and more than 80 times faster inference time than the  previous state-of-the-art method.

**Strengths:**

A novel and efficient diffusion sampling strategy is proposed, synergistically combining diffusion sampling and Krylov subspace methods.
It certifies that if the tangent space at a denoised sample by Tweedie’s formula forms a Krylov subspace,
then the Conjugate Gradient initialized with the denoised data ensures that the data consistency update remains in the tangent space.
This also eliminates the need to compute the manifold-constrained gradient.

**Weaknesses:**

1.The detailed process of M-step CG update is lacking.
2.In page 7, I think clarifying Equation (29) will help the reader understand the problem setting.
3.An important point of this paper is the negation of the necessity to compute the manifold-constrained gradient,
but there is a lack of elaboration on using Tweedie's formula to obtain the tangent space for gradient calculation.

**Questions:**

1.In page 3, " the m-th order Krylov subspace is defined by equation (2)", this "m-th" is not shown in equation (2).
2.For high-dimensional data, the computational complexity of the Conjugate Gradient (CG) algorithm is significant, and each step of inverse sampling requires solving using the CG algorithm.
Does this affect the model's efficiency?
3.Which has a higher computational complexity, this method or the classical accelerated diffusion models(e.g., DDIM)?

---

> ### Author Response · Authors · 2023-11-19
> **Reply to reviewer tJRQ**
>
> **W1. The detailed process of $M$-step CG update is lacking.
>
> **A.** For completeness, we included Algorithm 1 which is the precise algorithm used throughout the work.
>
> **W2. In page 7, I think clarifying Eq. (29) will help the reader understand the problem setting.**
>
> **A.** We included Fig. 3 as an illustration of the imaging physics, and added a pointer in the main text.
>
> **W3. An important point of this paper is the negation of the necessity to compute the manifold-constrained gradient, but there is a lack of elaboration on using Tweedie's formula to obtain the tangent space for gradient calculation.**
>
> **A.** Thank you for the suggestion. In the modified Appendix B.1., where we cover the preliminaries on diffusion models, we have also elaborated on the use of Tweedie’s formula and how it relates to diffusion models and our work.
>
> **Q1. In page 3, $m$-th is not shown in equation (2).**
>
> **A.** Thank you for spotting the typo. Fixed.
>
> **Q2,3. For high-dim data, the compute complexity of CG is significant. Does this affect model's efficiency? How does it compare to DDIM?**
>
> **A.** The computation cost for diffusion sampling is approximately linear to the neural function evaluation (NFE) count. This is the computation time needed for neural network forward pass dominates and far exceeds the time needed for applying CG, even for the large-scale inverse problems that we consider.
>
> For instance, removing the CG steps, DDS would be equivalent to unconditional DDIM sampling. For the case of CS-MRI where the dimension is (15 $\times$ 320 $\times$ 320), DDIM vs DDS sampling time is 13.50 vs 13.01 [s], showing that there exists only a small difference.
>
> Even when we consider 3D SV-CT reconstruction where we additionally solve the optimization problem with ADMM-CG, and the data dimension is very high (256 $\times$ 256 $\times$ 448), DDIM vs DDS sampling time is 21:07 vs. 16:01 [m]:[s]. While in this case, the difference is more pronounced than the multi-coil CS-MRI case, we still see that DDS only introduces a reasonable amount of computational overhead considering the scale of the problem that we are solving.

---

### Official Review · Reviewer_wm9K · 2023-10-31

**Soundness:** 3 good
**Presentation:** 3 good
**Contribution:** 3 good
**Rating:** 6
**Confidence:** 2

**Summary:**

In this paper, the authors aim to solve the inverse problem through diffusion model and propose the conjugate gradient (CG) method based on its connection to Krylov subspace. Specifically, in each time step, they update the noisy estimation based on the CG update and finally obtain the inverse results. Experiments show that the proposed method works well.

**Strengths:**

- Using CG method in each time step is reasonable and can mitigate the difficulty of solving the original problem.
- The results seem promissing.

**Weaknesses:**

- How to initialize $x_0$, from Gaussian noise?
- In equation (19), what is $A_t$ and $y_t$, are they the same with $A$ and $y$? If so, will this cause some bias or problem?
- For different steps, will $l$ in equation (22) be fixed? If so, how to get an optimal $l$?
- In table 1, what's the meaning of mask pattern?
- In table 2, it seems that with more steps, the performance drops. Any explanation?

**Questions:**

Please see above

---

> ### Author Response · Authors · 2023-11-19
> **Reply to reviewer wm9K**
>
> **W1. How do we initialize $\mathbf{x}_0$ from Gaussian noise?**
>
> **A.** We sample from random Gaussian noise, i.e. sampling $\mathbf{x}_T \sim \mathcal{N}(0, \mathbf{I})$ as in standard practice of diffusion models.
>
> **W2. In equation (19), what is $A_t$ and $y_t$? Are they same with $A$ and $y$? If so, will this cause some bias or problem?**
>
> **A.** There is no $A_t$ and $y_t$ in eq. (19). Please let us know which equation you are referring to. We would be happy to answer your question.
>
> **W3. For different steps, will $l$ in equation (22) be fixed? If so, how to get an optimal $l$?
>
> **A.** Yes. For all steps $l$ is fixed in our experiments, although this does not necessarily have to be the case. We opted for simplicity and let $l$ fixed. $l$ is a hyper-parameter that one could tune according to the circumstances (e.g. see Table 2). In order to find $l$, one could rely on heuristics, but we note that in practice, we achieve stable results for a wide range of $l$.
>
> **W4. In table 1, what's the meaning of mask pattern?**
>
> **A.** The mask pattern refers to the under-sampling pattern of Fourier space in compressed sensing MRI. Please see the first column of Figure 6 and 7 for examples of these “masks”. The white pixels are the *sampled* k-space points, where as the black pixels are the *unknown* k-space points that need imputation.
>
> **W5. In table 2, it seems that with more steps, the performance drops. Any explanation?**
>
> **A.** Our theoretical analysis builds on top of the assumption that the clean data manifold is an $l$-th order Krylov subspace. When the iteration $M$ exceeds this value of $l$, the sample might deviate from the tangent, and thereby incur errors.

---

### Official Review · Reviewer_Q4Mc · 2023-11-01

**Soundness:** 3 good
**Presentation:** 2 fair
**Contribution:** 3 good
**Rating:** 6
**Confidence:** 3

**Summary:**

This work proposes the Decomposed Diffusion Sampling (DDS) method as a Diffusion model-based Inverse problem Solvers (DIS) for inverse problems in medical imaging. The work is based on the observation that a diffusion posterior sampling (DPS) with the manifold constrained gradient (MCG)  is equivalent to one-step projected gradient on the tangent space at the “denoised" data by Tweedie’s formula, this work
provides multi-step update scheme on the tangent space using Krylov subspace methods. The experiments show that performing numerical optimization schemes on the denoised representation is superior to the previous methods of imposing DC. Further, the work devises a fast sampler based on DDIM that works well for both VE/VP settings. With extensive experiments on multi-coil MRI reconstruction and 3D CT reconstruction, it was shown that DDS achieves superior quality while being ≥ ×80 faster than the previous DIS.

**Strengths:**

The main strength of the work is the novel theoretic insight: the geometric interpretation that the diffusion posterior sampling (DPS) with the manifold constrained gradient (MCG)  is one-step projection to the tangent space of the clean data manifold, and using Conjugate Gradient type method, one can achieve multiple steps projection within the tangent space. All the theoretic proofs are given in details, all the lemmas are well formulated and clearly explained. The experimental results are convincing.

**Weaknesses:**

The work involves both Krylov space theory and diffusion model, it will be more helpful for general audience to give brief overview of both theories in the appendix. Especially. the key observation : DPS with MCG is a one-step projection to the tangent space of the clean data manifold. Furthermore, it needs more explanation for the problem : why Krylov space method can guarantee to stay in the tangent space.

**Questions:**

1. Do we need the  assumption that the clean data manifold is an affine subspace ? How about general curved manifold ?
2. Why Krylov space method guarantee to stay in the tangent space ?
3. Please elaborate the geometric view of diffusion model more. Is the diffusion process on the manifold or in the ambient Euclidean space ?

---

> ### Author Response · Authors · 2023-11-19
> **Reply to reviewer Q4Mc**
>
> **W1, Q2. Brief overview on diffusion and Krylov space theory is needed. Why do Krylov subspace methods guarantee the iterations to stay in the tangent space?**
>
> **A.** Thank you for the suggestion. Per the reviewer's request, we have included the preliminaries section in Appendix A, where we review both diffusion models and Krylov subspace methods.
>
> Please see the modified Appendix A.2. on the background of Krylov subspace methods. In essence, Krylov subspace methods can be thought of as repeated applications of the matrix-vector product with some matrix $A$, which is responsible for constructing the span of the Krylov subspace. For every increased iteration, we have an increased order of the subset. Due to the property that the $M$-th order Krylov space is a subspace of $L$-th order Krylov space, i.e. $\mathcal{K}_M \subset \mathcal{K}_L$ if $M \leq L$, Krylov subspace methods are guaranteed to stay on the tangent space if the iteration count $M$ does not exceed the value of $L$.
>
> **Q1. Do we need the assumption that the clean data manifold is an affine subspace? How about general curved manifold?**
>
> **A.** This assumption is required for diffusion models to act as the projector. That said, it should be noted that natural data manifolds can be well-approximated as locally affine manifolds in most cases, which makes our assumption practical. Note that the affine subspace assumption is widely used in the diffusion literature and has been used when 1) studying the possibility of the score estimation and distribution recovery [1], 2) showing the possibility of signal recovery, and most relevantly [2,3], 3) showing the geometrical view of the clean and noisy manifolds. [4]
>
> **Q3. Please elaborate the geometric view of diffusion model more. Is the diffusion process on the manifold or in the ambient Euclidean space?**
>
> **A.** We apologize for the confusion. The diffusion process is on the ambient Euclidean space, but we make assumptions that the natural data manifold exists on a manifold, which is a subset of the Euclidean ambient space.
>
>
> **References**
>
> [1] Chen, Minshuo, et al. "Score approximation, estimation and distribution recovery of diffusion models on low-dimensional data." ICML 2023.
>
> [2] Rout, Litu, et al. "Solving linear inverse problems provably via posterior sampling with latent diffusion models." NeurIPS 2023.
>
> [3] Rout, Litu, et al. "A theoretical justification for image inpainting using denoising diffusion probabilistic models." arXiv preprint arXiv:2302.01217 (2023).
>
> [4] Chung, Hyungjin, et al. "Improving diffusion models for inverse problems using manifold constraints." NeurIPS 2022.

---

### Author Response · Authors · 2023-11-19
**General response**

We would like to thank the reviewers for their constructive and thorough reviews. We are encouraged that the reviewers think that **the work offers novel theoretic insight** (Q4Mc), **method is reasonable and results are promising** (wm9K), **proposes a novel and efficient strategy** (tJRQ), and **the idea is novel, significant, and clear** (QCsq).

For point-to-point response, please refer to below.

---

### Meta-Review · Area_Chair_xEAd · 2023-12-08

**Metareview:**

This paper introduces a novel diffusion sampling strategy that combines diffusion sampling with Krylov subspace methods for solving inverse problems in medical imaging. The approach uses the fact that the tangent space at a denoised sample forms a Krylov subspace. A Conjugate Gradient (CG) method, starting with the denoised data, can be used to ensure updates in data consistency remain within the tangent space. This strategy bypass the need for computing the manifold-constrained gradient (MCG) -- this enhances efficiency. The method is versatile, applicable in various settings, and shoes significant improvements in reconstruction quality for complex medical imaging tasks (eg: multi-coil MRI and 3D CT reconstruction). It achieves these results with an inference time more than 80 times faster than recent competing methods.

**Justification For Why Not Higher Score:**

The reviewers noted that while the methodology has potential for addressing nonlinear problems, this aspect was not extensively explored in the text. Additionally, they mentioned that the paper's exposition was occasionally unclear and could benefit from improvement.

**Justification For Why Not Lower Score:**

The paper provides new insights, and the community will likely benefit from the ideas it presents.

---

### Decision · Program_Chairs · 2024-01-16

Accept (poster)